# Scaling Epidemic Inference on Contact Networks: Theory and Algorithms

**Guanghui Min**   **Yinhan He**   **Chen Chen**
University of Virginia
{jjm8vr, nee7ne, zrh6du}@virginia.edu

## Abstract

Computational epidemiology is crucial in understanding and controlling infectious diseases, as highlighted by large-scale outbreaks such as COVID-19. Given the inherent uncertainty and variability of disease spread, Monte Carlo (MC) simulations are widely used to predict infection peaks, estimate reproduction numbers, and evaluate the impact of non-pharmaceutical interventions (NPIs). While effective, MC-based methods require numerous runs to achieve statistically reliable estimates and variance, which suffer from high computational costs. In this work, we present a unified theoretical framework for analyzing disease spread dynamics on both directed and undirected contact networks, and propose an algorithm, **RAPID**, that significantly improves computational efficiency. Our contributions are threefold. First, we derive an asymptotic variance lower bound for MC estimates and identify the key factors influencing estimation variance. Second, we provide a theoretical analysis of the probabilistic disease spread process using linear approximations and derive the convergence conditions under non-reinfection epidemic models. Finally, we conduct extensive experiments on six real-world datasets, demonstrating our method's effectiveness and robustness in estimating the nodes' final state distribution. Specifically, our proposed method consistently produces accurate estimates aligned with results from a large number of MC simulations, while maintaining a runtime comparable to a single MC simulation. Our code and datasets are available at https://github.com/GuanghuiMin/RAPID.

## 1 Introduction

In recent years, outbreaks of infectious diseases such as COVID-19 have highlighted the critical need for accurate modeling and prediction of disease dynamics [15, 8, 5, 49]. Existing epidemic studies tend to model the disease spread process using differential equations at the population level, and often assume homogeneous populations with random mixing of initial infections within contact networks [27]. Consequently, such work can only provide a rough estimate of the overall dynamics of epidemics, while missing important details about individual- or local-level infection dynamics in the networks. In practice, individual-level infection dynamic inference has important practical implications. For example, individuals with a high probability of transmitting the disease can be prioritized for vaccination, thus reducing the overall infected population [22]. In addition, understanding the results of local infections can provide a more accurate projection of the global trajectory of the epidemic. Although some recent studies have enabled advances in understanding the interplay of network structure and epidemic spread process, they can only provide an insight about the epidemic at a high level by determining whether an outbreak will escalate into a complete pandemic [50, 16, 43], while neglecting the impact of distribution of initial cases and offering limited information on the infection states and expected size of the infected population.

39th Conference on Neural Information Processing Systems (NeurIPS 2025).

Currently, inferring the infectious state of individual nodes in the epidemic network heavily depends on the Monte Carlo (MC) simulations, as they do not rely on assumptions of population homogeneity or specific distributions of initial infections [60, 14, 10, 11, 21, 19]. For example, during the COVID-19 pandemic, MC simulations have been widely employed to predict infection peaks, estimate reproduction numbers, and assess the impact of non-pharmaceutical interventions (NPIs) [60, 14]. Due to the simplicity and robustness of MC simulations, they are widely adopted in various related application domains such as information dissemination and cybersecurity, where network-based contagion models play a crucial role [26, 47, 57]. However, a non-negligible drawback for MC simulations is their high computational cost, as it requires hundreds or even thousands of runs of simulations to reach a statistically reliable result due to the inherent uncertainty and variability of the disease spread process [43].

In this paper, we address the high computational cost of MC simulations for modeling the disease spread process by presenting an efficient framework to estimate the infection dynamics on large networks while ensuring high estimation accuracy on predicting the infection states of individual nodes at convergence. To our knowledge, we are the first to theoretically analyze how the network structure impacts the variance of epidemic simulations and provide a linear approximation scheme for nonlinear epidemic dynamics with provable performance guarantees. Our key contributions are as follows:

- **_Theoretical Analysis:_** We present three key results: (1) we derive the variance lower bound of the Monte Carlo estimator on large contact networks and reveal how the reproduction number, average out-degree, initial infection ratio, and the number of simulations affect the variance; (2) we provide a linear approximation of the nonlinear epidemic dynamics, clarifying the contribution of neighborhoods to disease spread; and (3) we establish global convergence conditions for probability-based propagation under non-reinfection epidemic models.

- **_Algorithm and Results:_** Building on these theoretical insights, we propose **RAPID** (**R**esidual-**A**ware **P**ropagation for **I**nfection **D**ynamics), a local approximation algorithm designed to efficiently estimate the node-level state distribution in large networks while preserving interpretability and reliability throughout the computational process. In particular, RAPID consistently produces accurate estimates aligned with results from a large number of MC simulations, while maintaining a runtime comparable to a single MC simulation.

## 2 Preliminary

**Notations.** We denote a general graph[1] as $\mathcal{G} = (\mathcal{V}, \mathcal{E}, \mathbf{A})$, where $\mathcal{V}$ is the set of nodes and $N = |\mathcal{V}|$ is the number of nodes; $\mathcal{E}$ denotes the set of edges, and the edge between nodes $v_i$ and $v_j$ is denoted as $e_{ij} = (v_i, v_j)$; $\mathbf{A} \in \{0, 1\}^{N \times N}$ is the adjacency matrix, where $\mathbf{A}_{ij} = 1$ indicates that an edge exists between $v_i$ and $v_j$, otherwise $\mathbf{A}_{ij} = 0$. In this work, we assume a standard SIR (Susceptible-Infected-Recovered) model with a constant transmission probability $\beta$ and recovery probability $\gamma$ governs the spread of a virus on this network. All notations and symbols are summarized in Table 1.

**Definition 1** (Individual-Level Epidemic Inference)**.** *Given a directed contact network $\mathcal{G} = (\mathcal{V}, \mathcal{E}, \mathbf{A})$, epidemic parameters $\beta$ and $\gamma$, and an initial infected node set $\mathcal{I}_0$, the goal of individual-level epidemic inference is to estimate the steady-state (converged) state probability distribution $P^i(\infty) = [P_S^i(\infty), P_I^i(\infty), P_R^i(\infty)]^T$ for each node $i \in \mathcal{V}$.*

This problem fundamentally differs from typical graph-based spatio-temporal prediction tasks, which aim to forecast future node-level observations (such as traffic or sensor signals) based on historical data. In contrast, node-level epidemic inference seeks to estimate the final distribution of susceptible probabilities at convergence with given initial conditions and epidemic parameters, and the contagion process follows the stochastic infection dynamics without relying on past observation sequences.

---

[1]In this work, the graph or network can be either directed or undirected. Without loss of generality, we derive our analysis on directed graphs as undirected graphs can be considered as a special case of directed graphs.

Table 1: Summary of key notations

| Graph | |
|---|---|
| $\mathcal{G} = (\mathcal{V}, \mathcal{E}, \mathbf{A})$ | Directed contact network with node set $\mathcal{V}$, edge set $\mathcal{E}$, and adjacency matrix $\mathbf{A}$. |
| $N$ | Number of nodes, $|\mathcal{V}|$. |
| $\bar{k}, D$ | Average out-degree, graph diameter. |
| $\mathcal{NE}^{\text{in}}(i), \mathcal{NE}^{\text{out}}(i)$ | In-/out-neighbors of node $i$. |
| **Epidemic parameters** | |
| $\beta, \gamma$ | Transmission probability, recovery probability in unit time. |
| $\mathcal{I}_0, \alpha$ | Initial infected node set, with fraction $\alpha = |\mathcal{I}_0|/N$. |
| $p_e$ | Edge-level transmission probability, $p_e = \beta/(\beta + \gamma)$. |
| **State variables** | |
| $P_S^i(t), P_I^i(t), P_R^i(t)$ | Probabilities that node $i$ is susceptible, infected, or recovered at time $t$. |
| $P^i(t)$ | Full state vector of node $i$: $[P_S^i(t), P_I^i(t), P_R^i(t)]^T$. |
| $P_S(t), P_I(t), P_R(t)$ | Vectors over all nodes, e.g., $P_S(t) = [P_S^1(t), \dots, P_S^N(t)]^T$. |
| $P(t)$ | Full network state: $[P^1(t), \dots, P^N(t)]^T$. |
| $R_{\text{res}}(i)$ | Propagation residual at node $i$. |

## 2.1 SIR Dynamics

The classical SIR model describes disease transmission as a continuous-time Markov process, governed by the following system of ordinary differential equations [27]:

$$\frac{dS}{dt} = -\beta \frac{SI}{N}, \quad \frac{dI}{dt} = \beta \frac{SI}{N} - \gamma I, \quad \frac{dR}{dt} = \gamma I. \tag{1}$$

These nonlinear equations are generally difficult to solve analytically, and the implicit assumption of homogeneous random mixing may not hold in complex contact networks. Notably, the SIR framework serves as a representative example of general non-reinfection epidemic models, where once nodes transition into the recovered (or removed) state, they no longer re-enter the infection dynamics [46, 28].

## 2.2 Probabilistic Disease Propagation in Neighborhood

Given a directed contact network $\mathcal{G} = (\mathcal{V}, \mathcal{E}, \mathbf{A})$, epidemic parameters $\beta$ and $\gamma$, and the full network state $P(t)$ at time $t$, we model the SIR process as a non-reinfection system. Specifically, for each node $i \in \mathcal{V}$, the probability of being susceptible at time $t+1$, denoted $P_S^i(t+1)$, is given by its value at time $t$ multiplied by the probability of not being infected during the interval $(t, t+1]$. The probability of being infected at time $t+1$, $P_I^i(t+1)$, consists of two parts: the probability of becoming infected during $(t, t+1]$ while being susceptible at $t$, and the probability

---

**Algorithm 1** Sketch of **PID**

**Require:** Graph $\mathcal{G}$, rates $\beta, \gamma$, initial infected set $\mathcal{I}_0$, threshold $\varepsilon$
1: Initialize $P_S, P_I, P_R, t$
2: **while** $\|P(t) - P(t-1)\|_2 > \varepsilon$ **do**
3:     **for all** $i \in \mathcal{V}$ **do**
4:         Update $P_S^i(t), P_I^i(t), P_R^i(t)$ with Eq. (2)
5:     **end for**
6:     $t \leftarrow t + 1$
7: **end while**
8: **return** $P_S, P_I, P_R$

---

of remaining infected by not transitioning to the recovered state. Similarly, the probability of being recovered at $t+1$, $P_R^i(t+1)$, accounts for nodes that either recovered during $(t, t+1]$ or remained in the recovered state from the previous time step. Assuming that infection events are independent across neighbors within each unit time interval [43, 25], the resulting update equations correspond to the derivation of the message passing formulation in [24]:

$$P_S^i(t+1) = P_S^i(t) \prod_{j \in \mathcal{V}} \mathbf{A}_{ji}(1 - \beta P_I^j(t)) \tag{2}$$

$$P_I^i(t+1) = P_S^i(t)[1 - \prod_{j \in \mathcal{V}} \mathbf{A}_{ji}(1 - \beta P_I^j(t))] + (1 - \gamma)P_I^i(t)$$

$$P_R^i(t+1) = P_R^i(t) + \gamma P_I^i(t)$$

Based on these update equations, we develop a baseline inference algorithm, **PID** (**P**ropagation for **I**nfection **D**ynamics), as shown in Algorithm 1.

# 3 Theoretical Analysis

In this section, we provide a thorough analysis on the impact of network structure and epidemic parameters on the variance of variance of Monte Carlo estimators; then we derive a linear approximation of nonlinear epidemic dynamics that highlights neighborhood propagation, and establish convergence guarantees for probability-based propagation in non-reinfection models.

## 3.1 Monte Carlo Estimator Variance Analysis

In many existing epidemic dynamic studies, Monte Carlo simulations are often repeated $10^4$–$10^5$ times without examining their necessity or computational trade-offs. In this subsection, we aim to theoretically investigate this important question: *under what conditions does the variance of the steady-state infection probability across the network become large when performing Monte Carlo simulations of epidemic dynamics, such that more simulation runs are needed to achieve sufficient accuracy?* To the best of our knowledge, this question has never been explored in prior work. Addressing this overlooked gap not only offers practical guidance for real-world simulation tasks but also motivates our development of scalable methods for epidemic inference.

**Theorem 3.1** (Monte Carlo Estimation Variance Lower Bound). *Given a directed contact network $\mathcal{G} = (\mathcal{V}, \mathcal{E})$ with $N = |\mathcal{V}|$, average out-degree $\bar{k}$, and diameter $D$. Let $\mathcal{I}_0 \subseteq \mathcal{V}$ be the initially infected node set with fraction $\alpha := |\mathcal{I}_0|/N$. Assume an SIR model parameterized by infection probability $\beta$ and recovery probability $\gamma$. Using $M$ independent Monte Carlo simulations to estimate each node's infection probability $p_i$, the average variance of the estimator $\hat{p}_i$ satisfies:*

$$\frac{1}{N} \sum_{i=1}^{N} \mathrm{Var}(\hat{p}_i - p_i) \gtrsim \frac{1}{2M} \min\{1 - (1 - p_0)^{c\bar{k}\alpha}, (1 - p_0)^{c\bar{k}\alpha}\}, \tag{3}$$

*where*

$$p_0 := (\frac{\beta}{\beta + \gamma})^\ell, \quad \ell := \min\{D, \frac{\log N}{\log \bar{k}}\}, \tag{4}$$

*and $c > 0$ is a constant depending on the network structure.*

*Proof.* We briefly outline the key argument here; the full derivation is in the Appendix. Each node's infection probability $p_i$ is estimated via Monte Carlo simulation as

$$\hat{p}_i = \frac{1}{M} \sum_{m=1}^{M} \mathbb{I}_i^{(m)}, \tag{5}$$

where $\mathbb{I}_i^{(m)}$ indicates infection in simulation $m$. Its variance is straightforwardly given by $\mathrm{Var}[\hat{p}_i] = p_i(1 - p_i)/M$. Under the continuous-time SIR model, the transmission probability along any edge is characterized by competing exponential infection and recovery processes, resulting in $p_e = \beta/(\beta+\gamma)$ [28, 46]. Assuming independence and sparse network structure, node-level infection probabilities are approximated as $p_i \approx 1 - \prod_{j=1}^{m_i}(1 - p_e^{\ell_{ij}})$, where $\ell_{ij}$ represents the path length between nodes $i$ and $j$ and we neglect higher-order interactions and overlapping paths [25, 43].

The number of disjoint infection paths $m_i$ scales approximately as $c\bar{k}\alpha$, reflecting the influence of average degree $\bar{k}$ and initial infection fraction $\alpha$ [43, 25, 28]. Typical path lengths $\ell$ can be approximated as $\ell = \min\{D, \log N/\log \bar{k}\}$ [9, 42], leading to the simplified expression $p_i \approx 1 - (1 - p_0)^{c\bar{k}\alpha}$, with $p_0 := (p_e)^\ell$. Given the concavity of $p_i(1 - p_i)$, we apply a standard lower bound, yielding $p_i(1 - p_i) \geq \frac{1}{2}\min\{p_i, 1 - p_i\}$. Averaging this lower bound across all nodes, we obtain the claimed variance lower bound for the Monte Carlo estimator:

$$\frac{1}{N} \sum_{i=1}^{N} \mathrm{Var}(\hat{p}_i - p_i) \gtrsim \frac{1}{2M} \min\{1 - (1 - p_0)^{c\bar{k}\alpha}, (1 - p_0)^{c\bar{k}\alpha}\}. \tag{6}$$

$\square$

**Remark 1.** *Through Theorem 3.1, we observe that the variance of the Monte Carlo estimator is primarily influenced by four factors: $\beta/\gamma$, $\bar{k}$, $\alpha$, and $M$. The base probability $p_0 := (\beta/(\beta + \gamma))^\ell$ increases with $\beta/\gamma$, since the ratio $\beta/(\beta + \gamma)$ grows monotonically. Meanwhile, $\ell := \min\{D, \log N/\log \bar{k}\}$ decreases as $\bar{k}$ increases, leading to an indirect increase of $p_0$ with $\bar{k}$.*

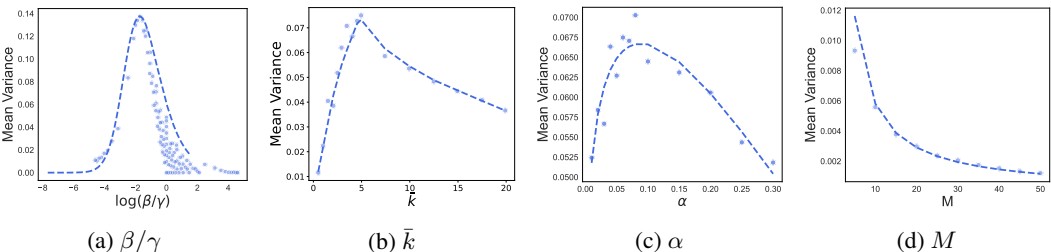

Figure 1: Influence of key factors on MC estimator variance.

As $p_0$, $\bar{k}$, or $\alpha$ increase, the term $(1 - p_0)^{c\bar{k}\alpha}$ in the variance bound decreases. Since the bound depends on $\min\{1 - (1 - p_0)^{c\bar{k}\alpha},\ (1 - p_0)^{c\bar{k}\alpha}\}$, it achieves its maximum when this term approaches $0.5$. Therefore, estimator variance increases initially with $\beta/\gamma$, $\bar{k}$, or $\alpha$, reaching a peak when the infection probability nears $0.5$, and then decreases as the infection probability saturates. In contrast, $M$ appears in the denominator and monotonically reduces $\bar{v}$ as $M$ increases. These factor effects are empirically verified in Figure 1.

## 3.2 Neighborhood Propagation Analysis

In Section 2.2, we show how diseases propagate probabilistically through local neighborhoods. However, using PID directly for inference can lead to great computational inefficiencies, especially on large-scale networks. A key bottleneck is the *path-level inefficiency*: in each update round, all nodes with non-zero infected probabilities must be updated, resulting in a propagation complexity of approximately $\mathcal{O}(|\mathcal{E}|)$ as more nodes become affected. In this section, we provide a theoretical analysis of neighborhood-level approximations and global convergence conditions, laying the groundwork for scalable inference methods.

**Remark 2.** *When using simulation-based methods for inference, it is common to approximate the continuous-time dynamics with a discrete-time process [18, 17]. Here, $\beta$ and $\gamma$ denote the per-unit-time transmission probability and healing probability, respectively. However, when $\beta$ and $\gamma$ take relatively large values, it becomes necessary to scale them down to smaller time units, i.e., setting $\beta := \beta\Delta_t$ and $\gamma := \gamma\Delta_t$. This adjustment ensures consistency with the continuous infection rate:*

$$\lim_{\Delta_t \to 0} 1 - (1 - \beta\Delta_t)^{1/\Delta_t} = \beta + \mathcal{O}(\beta^2), \tag{7}$$

*which approximates linear growth for small $\beta$. Otherwise, if $\beta$ and $\gamma$ are large, the non-linear higher-order terms become non-negligible and can introduce significant errors in inference. Therefore, in the following discussions, we assume scaled-down $\beta$ and $\gamma$ satisfying $\beta \ll 1$ and $\gamma \ll 1$.*

A major challenge in SIR inference is that the underlying dynamic system is highly nonlinear, as shown in Eq. (2). Remark 2 guarantees that the higher-order errors remain sufficiently small. Following the linearization technique of [47], we locally linearize the nonlinear dynamics and derive the following theorem.

**Theorem 3.2** (Linear Approximation of Infection Dynamics)**.** *Given a directed contact network $\mathcal{G} = (\mathcal{V}, \mathcal{E}, \mathbf{A})$. Assume an SIR model parameterized by infection probability $\beta$ and recovery probability $\gamma$. The infection probability update equation in Eq. (2)*

$$P_I^i(t+1) = P_S^i(t) - P_S^i(t+1) + (1 - \gamma)P_I^i(t) \tag{8}$$

*can be approximated by the following linear expression:*

$$P_I^i(t+1) \approx P_S^i(t)(\beta \sum_{j \in \mathcal{V}} \mathbf{A}_{ji} P_I^i(t)) + (1 - \gamma)P_I^i(t). \tag{9}$$

*Proof.* We sketch the main argument; detailed derivation is provided in the Appendix. Under the SIR model on a directed contact network $\mathcal{G} = (\mathcal{V}, \mathcal{E}, \mathbf{A})$, let $P_I^i(t)$ and $P_S^i(t)$ denote the marginal probabilities that node $i$ is infected or susceptible at time $t$. The infection update equation follows from accounting for new infections and recoveries in Eq. (2):

$$P_I^i(t+1) = P_S^i(t)[1 - \prod_{j \in \mathcal{V}} \mathbf{A}_{ji}(1 - \beta P_I^j(t))] + (1 - \gamma)P_I^i(t). \tag{10}$$

Here, $P_S^i(t) - P_S^i(t+1)$ represents the probability that node $i$ transitions from susceptible to infected at time $t+1$, while $(1-\gamma)P_I^i(t)$ reflects staying infected without recovery. We then apply the first-order approximation $\log(1-x) \approx -x$ for small $x$ to linearize the product term $\prod_j(1 - \beta\mathbf{A}_{ji}P_I^j(t))$. This yields the following approximation for the probability of remaining susceptible:

$$P_S^i(t+1) \approx P_S^i(t)(1 - \beta\sum_{j\in\mathcal{V}}\mathbf{A}_{ji}P_I^j(t)). \tag{11}$$

Substituting into the update equation, we obtain:

$$P_I^i(t+1) \approx P_S^i(t)(\beta\sum_{j\in\mathcal{V}}\mathbf{A}_{ji}P_I^j(t)) + (1-\gamma)P_I^i(t). \tag{12}$$

This linear form captures the cumulative infection pressure from neighbors and the retention of infection due to non-recovery with probability $1 - \gamma$. $\qquad\square$

**Theorem 3.3** (Convergence Condition for Non-Reinfection Epidemic Models). *Consider a generalized non-reinfection epidemic model (e.g., $S^*I^2V^*$ [50]) over a directed contact network $\mathcal{G} = (\mathcal{V}, \mathcal{E}, \mathbf{A})$, where each susceptible node becomes infected with probability $\beta$ upon contact with an infected neighbor, and each infected node transitions to an absorbing state with probability $\gamma$ at each time step. Then, the system's state distribution $P(t)$ converges if and only if*

$$\max_{i\in\mathcal{V}} P_I^i(t) \to 0. \tag{13}$$

*Proof.* We sketch our proof as follows; detailed derivations are provided in the Appendix. We divide the argument into necessity and sufficiency. For necessity, we observe that if the state difference $\|P(t) - P(t-1)\|_2$ tends to zero, then by the update equation of the absorbing state, $\|P_{V^*}(t) - P_{V^*}(t-1)\|_2 = \gamma\|P_I(t-1)\|_2$ must also vanish. Since this term is bounded by the total state difference, we immediately obtain $\|P_I(t-1)\|_2 \to 0$, and thus $\max_{i\in\mathcal{V}} P_I^i(t-1) \to 0$.

For sufficiency, assume $\max_{i\in\mathcal{V}} P_I^i(t-1) \to 0$, which implies that $\|P_I(t-1)\|_2 \to 0$. Since the absorbing state update satisfies $\|P_{V^*}(t) - P_{V^*}(t-1)\|_2 = \gamma\|P_I(t-1)\|_2$, it follows that $\|P_{V^*}(t) - P_{V^*}(t-1)\|_2 \to 0$. Moreover, we show in the Appendix that $\|P_I(t) - P_I(t-1)\|_2 \to 0$. By conservation of total probability, we then have:

$$\|P_{S^*}(t) - P_{S^*}(t-1)\|_2 \le \|P_I(t) - P_I(t-1)\|_2 + \|P_{V^*}(t) - P_{V^*}(t-1)\|_2. \tag{14}$$

Combining these bounds shows that all components in $\|P(t) - P(t-1)\|_2$ vanish as $t$ increases, proving the claim. $\qquad\square$

**Remark 3.** *Theorem 3.3 implies that under non-reinfection epidemics models such as SIR, SEIR, or SIRD, the process inevitably converges as the infected probability at every node vanishes asymptotically, i.e., $\forall i \in \mathcal{V}$, $P_I^i(t) \to 0$. Consequently, the entire disease dynamics can be equivalently analyzed as a transition from susceptible to non-susceptible states ($S^* \to V^*$), without needing to explicitly model the transient infection state in the long term.*

## 4 Methodology

In this section, we first introduce the proposed algorithm **RAPID**, and then analyze its computational complexity. Our design ensures high efficiency, while the theoretical results in [24, 37] guarantee the accuracy of inference on sparse or locally tree-like epidemic contact networks, which are commonly encountered in practice.

### 4.1 The Proposed Algorithms

As our theoretical analysis in section 3.2 suggested that PID may suffer from computational inefficiency in the neighborhood propagation process, we propose an accelerated counterpart, **RAPID** (**R**esidual-**A**ware **P**ropagation for **I**nfection **D**ynamics), to efficiently scale inference on large contact networks. Our key idea is to define a *propagation residual* that quantifies the potential change in a node's infection probability due to local propagation. Nodes with large residuals are marked as active, and infection updates are computed only for these nodes, significantly reducing the overall computational cost.

First, we analyze the source of a one-step update on a node's infection probability. According to Theorem 3.2, we have

$$P_I^i(t+1) \approx P_S^i(t)(\beta \sum_{j \in \mathcal{V}} \mathbf{A}_{ji} P_I^j(t)) + (1-\gamma)P_I^i(t). \tag{15}$$

Taking the difference between consecutive time steps, we get:

$$
\begin{aligned}
P_I^i(t+1) - P_I^i(t) &\approx P_S^i(t)(\beta \sum_{j \in \mathcal{V}} \mathbf{A}_{ji} P_I^j(t)) - P_S^i(t-1)(\beta \sum_{j \in \mathcal{V}} \mathbf{A}_{ji} P_I^j(t-1)) \\
&\quad + (1-\gamma)(P_I^i(t) - P_I^i(t-1)) \\
&\leq P_S^i(t-1)(\beta \sum_{j \in \mathcal{V}} \mathbf{A}_{ji}(P_I^j(t) - P_I^j(t-1))) + (1-\gamma)(P_I^i(t) - P_I^i(t-1)) \\
&\leq \underbrace{\beta \sum_{j \in \mathcal{V}} \mathbf{A}_{ji}(P_I^j(t) - P_I^j(t-1))}_{\text{Change due to propagation}} + \underbrace{(1-\gamma)(P_I^i(t) - P_I^i(t-1))}_{\text{Change due to retained infection}}.
\end{aligned}
\tag{16}
$$

We aim to update a node's state through local propagation only when the external influence is sufficiently strong. Based on the derivation in Eq. (16), we define

$$R_{\text{res}}(i) = \beta \sum_{j \in \mathcal{V}} \mathbf{A}_{ji} (P_I^j - \tilde{P}_I^j), \tag{17}$$

as the *propagation residual* at node $i$, where $P_I^j$ denotes the current infected probability of node $j$, and $\tilde{P}_I^j$ denotes its cached value before the most recent update.

An overview of **RAPID** is shown in Algorithm 2. We first perform $p$ rounds of probabilistic disease propagation using Eq.(2), which serves as a preheating phase to initialize node states and compute propagation residuals. After preheating, we maintain a max-heap priority queue (referred to as the active queue) sorted by node residuals. At each iteration, the node $i$ with the highest residual is popped and its full state vector $\vec{P}^i$ is updated using Eq.(2) in line 13.

The resulting change in infection probability, $P_I^i - \tilde{P}_I^i$, is linearly propagated to its out-neighbors by incrementing their propagation residuals. If a neighbor's updated residual exceeds the threshold, it is pushed into the active queue for future propagation as is shown from line 15 to line 19. After propagation, the residual of the current node is set to zero, indicating it is no longer active. The algorithm terminates when the active queue becomes empty. Since $R_{\text{res}}(i) \leq \beta \sum_{j \in \mathcal{V}} \mathbf{A}_{ji} P_I^j$, Theorem 3.3 guarantees that once the system converges ($\max_{i \in \mathcal{V}} P_I^i \to 0$), the residuals vanish ($R_{\text{res}}(i) \to 0$), ensuring convergence of **RAPID**.

---

**Algorithm 2** Sketch of **RAPID**

**Require:** Graph $\mathcal{G}$, threshold $\varepsilon$, rates $\beta, \gamma$, initial infected set $\mathcal{I}_0$, preheat steps $p$
1: Initialize $P_S, P_I, P_R$
2: Preheat system for $p$ rounds and record $\tilde{P}_I$
3: Initialize max heap $\mathcal{H} \leftarrow \emptyset$
4: **for** $v \in \mathcal{V}$ **do**
5:     Compute $(R_{\text{res}}(v)$ with Eq. (17)
6:     Push $(R_{\text{res}}(v), v)$ into $\mathcal{H}$
7: **end for**
8: **while** $\mathcal{H}$ not empty **do**
9:     $(R_{\text{res}}(i), i) \leftarrow \text{popTop}(\mathcal{H})$
10:     **if** $R_{\text{res}}(i) \leq \varepsilon$ **then break**
11:     **end if**
12:     **if** $P_I^i > \varepsilon$ **then**
13:         Update $P_S^i, P_I^i, P_R^i, \tilde{P}_I^i$
14:         Set $\delta \leftarrow P_I^i - \tilde{P}_I^i$
15:         **for** $j \in \mathcal{NE}^{\text{out}}(i)$ **do**
16:             $R_{\text{res}}(j) \leftarrow R_{\text{res}}(j) + \beta \cdot \delta$
17:             **if** $R_{\text{res}}(j) > \varepsilon$ **then** push $j$ into $\mathcal{H}$
18:             **end if**
19:         **end for**
20:     **end if**
21:     $R_{\text{res}}(i) \leftarrow 0$
22: **end while**
23: **return** $P_S, P_I, P_R$

---

## 4.2 Complexity Analysis

**Theorem 4.1** (Worst-Case Time Complexity of RAPID). *Consider* RAPID *on a directed contact network* $\mathcal{G} = (\mathcal{V}, \mathcal{E})$ *with* $N = |\mathcal{V}|$ *nodes and average out-degree* $\bar{k}$. *Let the infection probability be* $\beta$, *the recovery probability be* $\gamma$, *and the residual threshold be* $\varepsilon > 0$. *Then, under worst-case conditions, the total time complexity of* RAPID *is*

$$\mathcal{O}(N \cdot \bar{k} \cdot \min(\frac{1}{\varepsilon}, \frac{1}{\beta})). \tag{18}$$

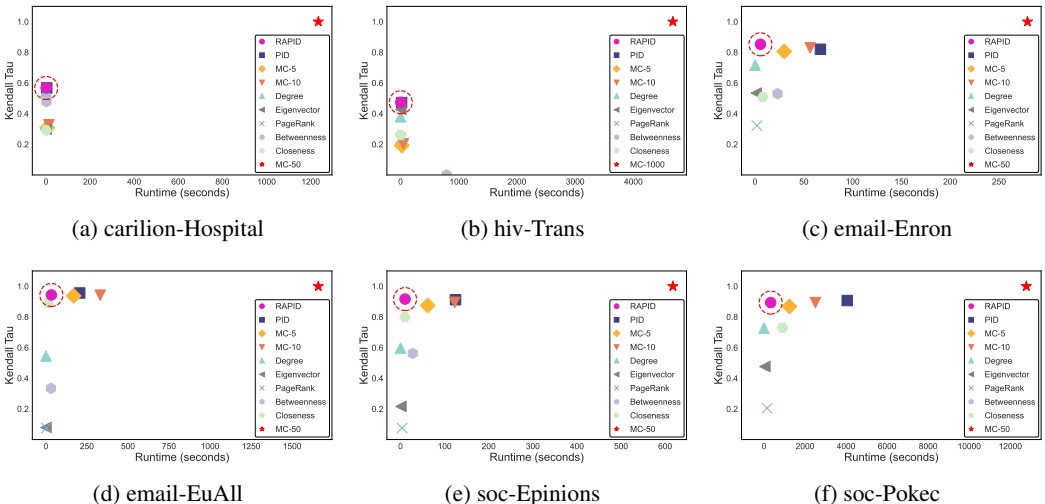

Figure 2: Trade-off between Kendall-Tau and Runtime across six datasets.

*Proof.* The key insight is that a node becomes active only if its residual exceeds $\varepsilon$, and the residual update it can receive in a single iteration approximates $\bar{k} \cdot \beta$, since infection can be passed through $\bar{k}$ neighbors on average at rate $\beta$. In the worst case, all $N$ nodes become active, and each may trigger residual pushes to up to $\bar{k}$ neighbors. When $\varepsilon$ is large (i.e., $\varepsilon \gg \bar{k}\beta$), only large residuals are propagated, and the number of total pushes is upper bounded by $\mathcal{O}(N \cdot \bar{k} \cdot \beta/\varepsilon)$. However, when $\varepsilon$ becomes very small (i.e., $\varepsilon \ll \bar{k}\beta$), further reducing $\varepsilon$ does not lead to more node activations, since the residuals themselves are bounded. Therefore, the total number of pushes saturates at $\mathcal{O}(N \cdot \bar{k})$. Combining both regimes gives the overall time complexity. This scaling behavior mirrors residual-based approximations in PPR push methods [3], but here it arises from the probabilistic infection dynamics of epidemic modeling. □

## 5 Empirical Study

In this section, we conduct extensive experiments to investigate the effectiveness and efficiency of the proposed method.

### 5.1 Experiment Settings

**Datasets.** We use graphs from diverse domains, including a real-world hospital contact network (`carilion-Hospital` [1]), a real-world HIV transmission network (`hiv-Trans` [40]), communication networks (`email-Enron` [35, 29], `email-EuAll` [34]), and social networks (`soc-Epinions` [52], `soc-Pokec` [54]). Among them, `carilion-Hospital` and `email-Enron` are undirected, while the others are directed. Nodes with no incoming or outgoing edges are removed and each undirected edge is represented as two reciprocal directed edges. Detailed statistics are summarized in Table 2. We also inclue a Bitcoin trade network (`bitcoin-Alpha` [33, 32]) in the appendix.

**Baselines.** To serve as baselines, we implemented a broad range of inference models. A detailed description is provided in the Appendix.

- *Network centrality measures:* Degree, Eigenvector [44], PageRank [45], Betweenness [6] and Closeness [53]. These methods assess structural importance and are evaluated by ranking.
- *MC-based inference:* 5-run and 10-run Monte Carlo simulations (MC-5 and MC-10).
- *Probability-based inference:* Propagation for Infection Dynamics (PID), which retains all propagation steps during disease spread process, and is an unaccelerated invariant of **RAPID**.

Table 2: Datasets Statistics

| Dataset | #Nodes | #Edges | Avg Deg |
|---|---|---|---|
| carilion-Hospital | 11,810 | 30,994 | 2.62 |
| hiv-Trans | 35,230 | 34,243 | 0.97 |
| email-Enron | 36,692 | 367,662 | 10.02 |
| email-EuAll | 265,214 | 420,045 | 1.58 |
| soc-Epinions | 75,879 | 508,837 | 6.71 |
| soc-Pokec | 1,632,803 | 30,622,564 | 18.75 |

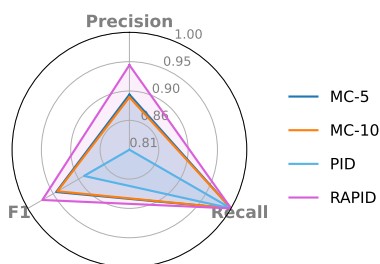

Figure 3: Classification metrics (precision, recall, F1) on `hiv-Trans`.

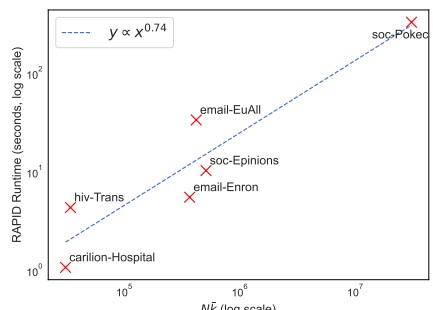

Figure 4: **RAPID** runtime grows sublinearly with graph size and density.

**Parameter Settings.** To ensure the practical relevance of our study, we consulted the literature to obtain the infectious period and basic reproduction number ($R_0$) for various diseases, as summarized in the Appendix. In our experiments, the ground-truth infection probability for each node is estimated by averaging its infection frequency over 50-run Monte Carlo simulations[2]. For reporting and baseline comparisons, we adopt the parameter setting of Nipah Virus ($\beta = 1/18$, $\gamma = 1/9$) with an initial infection fraction of $\alpha = 0.01$. Results for other disease parameters are similar and are provided in the Appendix.

## 5.2 Effectiveness and Efficiency Analysis

**Effectiveness of RAPID.** We evaluate the effectiveness of **RAPID** and baseline methods from three perspectives: the ranking of node-level infection probabilities, the classification performance of nodes whether infected, and the accuracy of infection probability estimation.

Figure 2 shows the trade-off between Kendall-$\tau$ and runtime across six datasets. **RAPID** achieves the best or near-best rank correlation on all of them while keeping the runtime consistently fast. We treat nodes with inferred infection probability above $0.5$ as infected. Figure 3 presents the classification results on `hiv-Trans`, where **RAPID** significantly improves precision over baselines, achieving the best overall performance.

Table 3 reports the mean absolute error (MAE) for infection probability estimation. **RAPID** consistently outperforms MC-5, MC-10, and PID across all datasets. As noted in [37], when contact networks contain many short loops, nodes within these loops cyclically influence each other's states over time, which reduces the accuracy of the original PID algorithm. **RAPID** addresses this issue by em-

Table 3: MAE comparison (lower is better). All values are scaled by $10^{-2}$. Best results are **in bold**.

| Dataset | MC-5 | MC-10 | PID | **RAPID** |
|---|---|---|---|---|
| carilion-Hospital | 13.01±0.35 | 10.78±0.44 | 5.67±0.51 | **2.64**±0.49 |
| hiv-Trans | 5.12±0.12 | 3.24±0.11 | 6.43±0.52 | **1.27**±0.47 |
| email-Enron | 7.60±0.02 | 5.98±0.03 | 8.70±0.04 | **4.66**±0.00 |
| email-EuAll | 2.09±0.01 | 1.63±0.01 | 1.36±0.02 | **1.03**±0.01 |
| soc-Epinions | 5.48±0.03 | 4.31±0.02 | 4.92±0.03 | **2.77**±0.00 |
| soc-Pokec | 4.50±0.00 | 3.54±0.00 | 3.32±0.00 | **2.32**±0.00 |

ploying asynchronous max-heap iterations to accelerate convergence and selective updates to mitigate the effect of short loops, thereby achieving higher inference accuracy than PID. Notably, although the Kendall-$\tau$ on `hiv-Trans` is lower than on other datasets, largely due to the large number of uninfected nodes at convergence, **RAPID** still achieves superior classification accuracy and lower MAE compared to all baselines.

**Runtime Comparison.** Table 4 shows the runtime of **RAPID** and baselines on each dataset. On the six datasets, **RAPID** achieves an average speedup of $5.11\times$, $10.67\times$, and $8.52\times$ over MC-5, MC-10, and PID, respectively. Compared to PID, the speedup is more pronounced on denser graphs, reaching $12.91\times$ on `soc-Epinions`. Against MC-10, **RAPID** exhibits stable acceleration ranging from $7.64\times$ to $12.58\times$, consistently maintaining a runtime comparable to that of single-run MC across all datasets.

---

[2]On `carilion-Hospital` and `hiv-Trans`, we adopt 1000-run MC simulations as the ground truth for acceptable estimator variance.

Table 4: Runtime comparison across datasets (seconds, lower is better). $\Delta$ indicates the speedup factor relative to RAPID, computed as $\Delta = \frac{\text{Baseline time}}{\text{RAPID time}}$.

| | | carilion-Hospital[2] | hiv-Trans[2] | email-Enron | email-EuAll | soc-Epinions | soc-Pokec |
|---|---|---|---|---|---|---|---|
| MC-5 | $t$ | $5.81_{\pm 0.51}$ | $21.87_{\pm 3.15}$ | $29.84_{\pm 1.43}$ | $169.46_{\pm 8.23}$ | $59.69_{\pm 1.26}$ | $1241.00_{\pm 18.79}$ |
| | $\Delta$ | $5.43\times$ | $5.16\times$ | $5.41\times$ | $5.06\times$ | $5.82\times$ | $3.78\times$ |
| MC-10 | $t$ | $13.46_{\pm 1.14}$ | $49.94_{\pm 1.74}$ | $56.45_{\pm 0.79}$ | $330.97_{\pm 6.81}$ | $122.31_{\pm 2.44}$ | $2506.60_{\pm 45.46}$ |
| | $\Delta$ | $12.58\times$ | $11.78\times$ | $10.24\times$ | $9.88\times$ | $11.91\times$ | $7.64\times$ |
| MC-50 | $t$ | $1234.73_{\pm 13.13}$ | $4678.18_{\pm 8.96}$ | $279.26_{\pm 3.09}$ | $1659.57_{\pm 23.55}$ | $614.58_{\pm 2.36}$ | $12782.37_{\pm 237.30}$ |
| | $\Delta$ | $1153.95\times$ | $1103.34\times$ | $50.66\times$ | $49.52\times$ | $59.86\times$ | $38.93\times$ |
| PID | $t$ | $3.56_{\pm 0.01}$ | $17.91_{\pm 0.14}$ | $66.95_{\pm 0.29}$ | $206.18_{\pm 0.65}$ | $132.60_{\pm 0.62}$ | $4056.89_{\pm 6.40}$ |
| | $\Delta$ | $3.33\times$ | $4.22\times$ | $12.14\times$ | $6.15\times$ | $12.91\times$ | $12.36\times$ |
| **RAPID** | $t$ | $\mathbf{1.07}_{\pm 0.00}$ | $\mathbf{4.24}_{\pm 0.03}$ | $\mathbf{5.51}_{\pm 0.04}$ | $\mathbf{33.50}_{\pm 0.05}$ | $\mathbf{10.27}_{\pm 0.09}$ | $\mathbf{328.28}_{\pm 0.66}$ |

**Complexity Validation.** Figure 4 shows the empirical log-scale runtime of **RAPID** versus $\log(N\bar{k})$ on six real-world networks. The fitted complexity runtime $\propto (N\bar{k})^{0.74}$ is better than the worst-case bound established in Theorem 4.1, confirming the validity of our analysis.

## 6   Conclusion

In this paper, we systematically studied the variance of Monte Carlo simulations for modeling the disease spread process in contact networks, and introduce a linear approximation for infection propagation under non-reinfection models with a provable convergence guarantee. Based on the theoretical findings, we propose **RAPID**, a residual-driven framework to infer node-level infection probability distribution with high estimation accuracy and low computational cost. Experiments on six real-world networks show that **RAPID** matches the accuracy of multi-run Monte Carlo within the runtime of a single simulation. Future work includes extensions to models with reinfection, time-varying parameters, and dynamic networks.

## Acknowledgments

The authors would like to thank the anonymous reviewers for their constructive comments. This work was supported in part by the NVIDIA Academic Grant Program, the Commonwealth Cyber Initiative (CCI) under Award No. HV-2Q25-032, and the National Science Foundation under Grant No. 2331315. Any opinions, findings, and conclusions or recommendations expressed in this material are those of the authors and do not necessarily reflect the views of the National Science Foundation.

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

# Supplemental Material

**Organization.** This supplementary material is organized as follows. Appendix A reviews related work and relevant theoretical background. Appendix B presents detailed proofs of the main theorems. Appendix C provides complete algorithm descriptions and highlights our design rationale from a **Jacobian perturbation** perspective, from which we derive the **classical epidemic threshold** and offer a physical interpretation of the propagation residual. Appendix D outlines implementation details and presents additional experiments, including: (i) how **Monte Carlo variance affects the reliability of MC-based inference**, (ii) the **extension of our method to the SEIR model**, (iii) the **impact of initial infection distributions**, (iv) experiments under alternative disease settings, and (v) a **sensitivity analysis of hyperparameters**. Appendix E discusses current limitations and potential directions for future research.

## A    Related Work

**Belief Propagation on Graphs.** Belief propagation (BP), also known as the sum–product algorithm, is a foundational message-passing technique for probabilistic inference on graphs. Originally introduced by Pearl for exact inference in tree-structured Bayesian networks [48], BP was later applied to graphs with cycles as an approximate method, i.e. loopy BP [41]. The algorithm iteratively transmits local "messages" along edges of a factor graph to compute marginal probability estimates [30]. Despite the lack of general convergence guarantees, loopy BP often produces accurate results and has achieved remarkable success across domains (e.g. decoding of error-correcting codes and vision networks). Numerous extensions have enhanced BP's accuracy and robustness. For instance, Yedidia et al. reinterpreted BP in terms of variational free-energy minimization and proposed generalized BP for higher-order region graphs [61]. Other work has focused on improving BP's convergence, such as methods that dampen oscillations or prioritize message updates based on residual magnitude [13]. These advances have solidified BP as a versatile approximate inference framework in graphical models.

**Local Push Algorithms and Variants.** LocalPush algorithms are a class of residual-based methods for approximate graph propagation, widely used in computing node influence scores or similarities on large graphs. A prime example is the estimation of Personalized PageRank (PPR) vectors, which measure the influence of a source node across a graph. Early approaches [23] found that by maintaining a "residual" at each node and pushing this probability mass outwards along edges, one can obtain accurate PPR estimates efficiently without full graph traversal. Andersen et al. formalized this idea by introducing a local push procedure with rigorous error bounds, enabling fast local graph partitioning via PageRank [4]. The LocalPush strategy restricts computation to the source's local neighborhood, yielding near-linear time updates for sparse networks. Subsequent research has built on this foundation to handle more complex settings. Notably, bidirectional and multi-target push algorithms improve efficiency for queries between specific node pairs [36]. Dynamic extensions have been developed to maintain PPR results under streaming graph updates, by locally adjusting residuals after each edge insertion or deletion [62]. Further optimizations address weighted graphs and precision guarantees, as in the edge-based LocalPush variant that decomposes pushes per edge to reduce redundant work [56]. Collectively, these residual-based propagation techniques have become essential tools for scalable graph mining, enabling approximate inference of importance or similarity scores with provable accuracy on massive networks.

## B    Theoretical Proof

### B.1    Proof of Theorem 3.1

**Theorem 3.1** (Monte Carlo Estimation Variance Lower Bound) Given a directed contact network $\mathcal{G} = (\mathcal{V}, \mathcal{E})$ with $N = |\mathcal{V}|$, average out-degree $\bar{k}$, and diameter $D$. Let $\mathcal{I}_0 \subseteq \mathcal{V}$ be the initially infected node set with fraction $\alpha := |\mathcal{I}_0|/N$. Assume an SIR model parameterized by infection probability $\beta$ and recovery probability $\gamma$. Using $M$ independent Monte Carlo simulations to estimate each node's

infection probability $p_i$, the average variance of the estimator $\hat{p}_i$ satisfies:

$$\frac{1}{N}\sum_{i=1}^{N}\mathrm{Var}(\hat{p}_i - p_i) \gtrsim \frac{1}{2M}\min\{1 - (1-p_0)^{c\bar{k}\alpha}, (1-p_0)^{c\bar{k}\alpha}\}, \tag{19}$$

where

$$p_0 := (\frac{\beta}{\beta+\gamma})^\ell, \quad \ell := \min\{D, \frac{\log N}{\log \bar{k}}\}, \tag{20}$$

and $c > 0$ is a constant depending on the network structure.

*Proof.* Each node's infection probability $p_i$ is estimated by Monte Carlo simulation:

$$\hat{p}_i := \frac{1}{M}\sum_{m=1}^{M}\mathbb{I}_i^{(m)}, \tag{21}$$

where $\mathbb{I}_i^{(m)} \in \{0, 1\}$ indicates whether node $i$ is infected in simulation $m$. The variance is:

$$\mathrm{Var}[\hat{p}_i] = \frac{1}{M}p_i(1-p_i). \tag{22}$$

Under the continuous-time SIR model, infection along an edge competes with recovery via independent exponential clocks. The transmission probability per edge is [28, 46]:

$$p_e := \Pr[T_{\mathrm{infect}} < T_{\mathrm{recover}}] = \frac{\beta}{\beta+\gamma}. \tag{23}$$

Assuming that infection is mainly driven by multiple independent disjoint paths [25, 43], the infection probability of node $i$ can be approximated as:

$$p_i \approx 1 - \prod_{j=1}^{m_i}(1 - p_e^{\ell_{ij}}), \tag{24}$$

where we neglect overlapping paths and higher-order interactions in sparse networks.

In sparse graphs, the number of disjoint infection paths to a node is heuristically assumed to scale with the average degree $\bar{k}$ and the initial infection fraction $\alpha$, yielding $m_i \approx c\bar{k}\alpha$. This scaling reflects the intuition that a higher initial infection fraction increases the number of potential sources and thus the likelihood of multiple disjoint infection paths [43, 25, 28]. The typical path length approximates $\ell := \min\{D, \log N/\log \bar{k}\}$ [9, 42]. Substituting these yields:

$$p_i \approx 1 - (1 - p_0)^{c\bar{k}\alpha}, \quad p_0 := (p_e)^\ell. \tag{25}$$

Since $p_i(1 - p_i)$ is concave with maximum at $p_i = \frac{1}{2}$, we lower bound it by:

$$p_i(1 - p_i) \geq \frac{1}{2}\min\{p_i, 1 - p_i\}. \tag{26}$$

Averaging this bound over all nodes (the bound is node-independent and thus unchanged by averaging), we obtain the claimed variance lower bound for the Monte Carlo estimator:

$$\frac{1}{N}\sum_{i=1}^{N}\mathrm{Var}(\hat{p}_i - p_i) \gtrsim \frac{1}{2M}\min\left\{1 - (1-p_0)^{c\bar{k}\alpha}, (1-p_0)^{c\bar{k}\alpha}\right\}. \tag{27}$$

$\square$

## B.2 Proof of Theorem 3.2

**Theorem 3.2** (Linear Approximation of Infection Dynamics) Given a directed contact network $\mathcal{G} = (\mathcal{V}, \mathcal{E}, \mathbf{A})$. Assume an SIR model parameterized by infection probability $\beta$ and recovery probability $\gamma$. The infection probability update equation

$$P_I^i(t+1) = P_S^i(t) - P_S^i(t+1) + (1-\gamma)P_I^i(t) \tag{28}$$

can be approximated by the following linear expression:

$$P_I^i(t+1) \approx P_S^i(t)(\beta\sum_{j\in\mathcal{V}}\mathbf{A}_{ji}P_I^i(t)) + (1-\gamma)P_I^i(t). \tag{29}$$

*Proof.* Substitute the susceptible update:

$$P_S^i(t+1) := P_S^i(t) \prod_j (1 - \beta \mathbf{A}_{ji} P_I^j(t)) \tag{30}$$

into

$$P_I^i(t+1) = P_S^i(t) - P_S^i(t+1) + (1-\gamma)P_I^i(t). \tag{31}$$

Take logarithm:

$$\log \left( \prod_j (1 - \beta \mathbf{A}_{ji} P_I^j(t)) \right) = \sum_j \log(1 - \beta \mathbf{A}_{ji} P_I^j(t)). \tag{32}$$

Approximate $\log(1 - x) \approx -x$ and exponentiate both sides:

$$\prod_j (1 - \beta \mathbf{A}_{ji} P_I^j(t)) \approx e^{-\beta \sum_j \mathbf{A}_{j,i} P_I^j(t)}. \tag{33}$$

According to Remark 2, we have $\beta \ll 1$. Use $1 - e^{-x} \approx x$:

$$1 - e^{-\beta \sum_j \mathbf{A}_{ji} P_I^j(t)} \approx \beta \sum_j \mathbf{A}_{ji} P_I^j(t). \tag{34}$$

Substitute into $P_I^i(t+1)$ to obtain the linear form. $\square$

### B.3 Proof of Theorem 3.3

**Lemma B.1.** $\|P_I(t-1)\|_2 \to 0$ *is sufficient for* $\|P_I(t) - P_I(t-1)\|_2 \to 0$.

*Proof.* This problem is equivalent to proving that for all $\varepsilon > 0$, there exists $\delta > 0$ such that if $\|P_I(t-1)\|_2 < \varepsilon$, then $\|P_I(t) - P_I(t-1)\|_2 < \delta$.

We have:

$$\max_{i \in \mathcal{V}} P_I^i(t-1) \le \|P_I(t-1)\|_2 \le \varepsilon. \tag{35}$$

First, for the absorbing state $P_{V^*}(t)$:

$$\|P_{V^*}(t) - P_{V^*}(t-1)\|_2 = \|\gamma P_I(t-1)\|_2 = \gamma \|P_I(t-1)\|_2 \le \gamma \varepsilon. \tag{36}$$

Next, for $P_S(t)$:

$$\begin{aligned}
P_S^i(t-1) - P_S^i(t) &= P_S^i(t-1) - P_S^i(t-1) \prod_{j \in \mathcal{V}} (1 - \beta \mathbf{A}_{j,i} P_I^j(t-1)) \\
&\le 1 - \prod_{j \in \mathcal{V}} (1 - \beta \mathbf{A}_{j,i} P_I^j(t-1)) \\
&\le 1 - (1 - \beta \varepsilon)^{d_{in}(i)}.
\end{aligned}$$

By Bernoulli's inequality, if $0 < x < 1$, then $(1-x)^{d_{in}(i)} \ge e^{-d_{in}(i)x}$. Letting $x = \beta \varepsilon$, we get:

$$(1 - \beta \varepsilon)^{d_{in}(i)} \ge e^{-d_{in}(i)\beta \varepsilon}. \tag{37}$$

Thus:

$$P_S^i(t-1) - P_S^i(t) \le 1 - e^{-d_{in}(i)\beta \varepsilon}. \tag{38}$$

Since $P_S^i(t-1) - P_S^i(t) \in (0,1)$, we can further bound:

$$1 - e^{-d_{in}(i)\beta \varepsilon} \le (1 - e^{-N\beta})\varepsilon. \tag{39}$$

Therefore:

$$P_S^i(t-1) - P_S^i(t) \le (1 - e^{-N\beta})\varepsilon. \tag{40}$$

Aggregating:

$$\|P_S(t-1) - P_S(t)\|_2 \le \sqrt{N}(1 - e^{-N\beta})\varepsilon. \tag{41}$$

Finally, for $P_I(t) - P_I(t-1)$:

$$\begin{aligned}
\|P_I(t) - P_I(t-1)\|_2 &= \|1 - P_S(t) - P_{V^*}(t) - (1 - P_S(t-1) - P_{V^*}(t-1))\|_2 \\
&= \|P_S(t-1) - P_S(t) + P_{V^*}(t-1) - P_{V^*}(t)\|_2 \\
&\leq \|P_S(t-1) - P_S(t)\|_2 + \|P_{V^*}(t) - P_{V^*}(t-1)\|_2 \\
&\leq \sqrt{N}(1 - e^{-N\beta})\varepsilon + \gamma\varepsilon.
\end{aligned}$$

Let:

$$\delta := \left(\sqrt{N}(1 - e^{-N\beta}) + \gamma\right)\varepsilon. \tag{42}$$

Then from $\max_{i \in \mathcal{V}} P_I^i(t-1) \leq \varepsilon$, we conclude:

$$\|P_I(t) - P_I(t-1)\|_2 \leq \delta. \tag{43}$$

$\square$

**Theorem 3.3** (Convergence Condition for Non-Reinfection Epidemic Models) Consider a generalized non-reinfection epidemic model (e.g., $S^*I^2V^*$ [50]) over a directed contact network $\mathcal{G} = (\mathcal{V}, \mathcal{E}, \mathbf{A})$, where each susceptible node becomes infected with probability $\beta$ upon contact with an infected neighbor, and each infected node transitions to an absorbing state with probability $\gamma$ at each time step. Then, the system's state distribution $P(t)$ converges if and only if

$$\max_{i \in \mathcal{V}} P_I^i(t) \to 0. \tag{44}$$

*Proof.* We first prove necessity. We aim to show that for any $\varepsilon > 0$, if $\|P(t) - P(t-1)\|_2 \leq \varepsilon$, then $\exists \delta > 0$ such that $\|P_I(t-1)\|_2 \leq \delta$. Since

$$\|P_{V^*}(t) - P_{V^*}(t-1)\|_2 = \gamma\|P_I(t-1)\|_2 \leq \|P(t) - P(t-1)\|_2 \leq \varepsilon, \tag{45}$$

it follows that

$$\|P_I(t-1)\|_2 \leq \frac{\varepsilon}{\gamma}. \tag{46}$$

Therefore,

$$\max_{i \in \mathcal{V}} P_I^i(t-1) \leq \|P_I(t-1)\|_2 \leq \delta := \frac{\varepsilon}{\gamma}. \tag{47}$$

Next we prove sufficiency. For any $\varepsilon > 0$, assume $\max_{i \in \mathcal{V}} P_I^i(t-1) \leq \varepsilon$. Then

$$\|P_I(t-1)\|_2 \leq \sqrt{N}\max_{i \in \mathcal{V}} P_I^i(t-1) \leq \sqrt{N}\varepsilon. \tag{48}$$

From the update of $P_{V^*}$,

$$\|P_{V^*}(t) - P_{V^*}(t-1)\|_2 = \|\gamma P_I(t-1)\|_2 = \gamma\|P_I(t-1)\|_2 \leq \gamma\sqrt{N}\varepsilon. \tag{49}$$

By Lemma B.1, since $\max_i P_I^i(t-1) \leq \varepsilon$, there exists $\delta' > 0$ such that $\|P_I(t) - P_I(t-1)\|_2 \leq \delta'$.
Since $P_S(t) + P_I(t) + P_{V^*}(t) = \mathbf{1}$,

$$\begin{aligned}
\|P_S(t) - P_S(t-1)\|_2 &= \|1 - P_I(t) - P_{V^*}(t) - (1 - P_I(t-1) - P_{V^*}(t-1))\|_2 \\
&= \|P_I(t-1) - P_I(t) + P_{V^*}(t-1) - P_{V^*}(t)\|_2 \\
&\leq \|P_I(t) - P_I(t-1)\|_2 + \|P_{V^*}(t) - P_{V^*}(t-1)\|_2 \\
&\leq \delta' + \gamma\sqrt{N}\varepsilon.
\end{aligned}$$

Therefore,

$$\begin{aligned}
\|P(t) - P(t-1)\|_2^2 &= \|P_S(t) - P_S(t-1)\|_2^2 + \|P_I(t) - P_I(t-1)\|_2^2 + \|P_{V^*}(t) - P_{V^*}(t-1)\|_2^2 \\
&\leq (\delta' + \gamma\sqrt{N}\varepsilon)^2 + (\delta')^2 + (\gamma\sqrt{N}\varepsilon)^2.
\end{aligned}$$

Taking square roots:

$$\|P(t) - P(t-1)\|_2 \leq \sqrt{(\delta' + \gamma\sqrt{N}\varepsilon)^2 + (\delta')^2 + (\gamma\sqrt{N}\varepsilon)^2} := \delta. \tag{50}$$

Thus, for any $\varepsilon > 0$, $\|P_I(t-1)\|_2 \leq \varepsilon$ implies $\|P(t) - P(t-1)\|_2 \leq \delta$. $\square$

# C Details of Algorithms

## C.1 Baselines

All centrality-based baselines (including Degree, Eigenvector [44], PageRank [45], Betweenness [6] and Closeness [53]) are implemented using Python's `networkx`[3] library.

## C.2 MC for SIR

---

**Algorithm 1: Monte Carlo Simulation for SIR**

---

**Require:** Graph $\mathcal{G} = (\mathcal{V}, \mathcal{E})$, infection rate $\beta$, recovery rate $\gamma$, initial infected set $\mathcal{I}_0$, number of trials $M$
**Ensure:** Estimated final-state probabilities $P_S^i, P_I^i, P_R^i$ for all $i \in \mathcal{V}$
1:                                                            ▷ Initialize counters
2:  $\text{count}_S^i \leftarrow 0, \quad \text{count}_I^i \leftarrow 0, \quad \text{count}_R^i \leftarrow 0$ for all $i \in \mathcal{V}$
3:  **for** $m = 1$ to $M$ **do**                                             ▷ Run $M$ independent trials
4:      $state^i \leftarrow S$ for all $i \in \mathcal{V}$
5:      **for** $i \in \mathcal{I}_0$ **do** $state^i \leftarrow I$
6:      **end for**
7:      **while** some node $i$ has $state^i = I$ **do**
8:          Initialize `next_state`$^i \leftarrow state^i$ for all $i$
9:          **for all** $i \in \mathcal{V}$ **do**
10:              **if** $state^i = I$ **then**
11:                  **for all** $j \in \mathcal{NE}^{\text{out}}(i)$ **do**
12:                      **if** $state^j = S$ and `rand()` $< \beta$ **then**
13:                          `next_state`$^j \leftarrow I$
14:                      **end if**
15:                  **end for**
16:                  **if** `rand()` $< \gamma$ **then**
17:                      `next_state`$^i \leftarrow R$
18:                  **end if**
19:              **end if**
20:          **end for**
21:          $state^i \leftarrow$ `next_state`$^i$ for all $i$
22:      **end while**
23:                                               ▷ Update final counts
24:      **for all** $i \in \mathcal{V}$ **do**
25:          **if** $state^i = S$ **then**
26:              $\text{count}_S^i \leftarrow \text{count}_S^i + 1$
27:          **end if**
28:          **if** $state^i = I$ **then**
29:              $\text{count}_I^i \leftarrow \text{count}_I^i + 1$
30:          **end if**
31:          **if** $state^i = R$ **then**
32:              $\text{count}_R^i \leftarrow \text{count}_R^i + 1$
33:          **end if**
34:      **end for**
35: **end for**
36:                                         ▷ Normalize to obtain final-state probabilities
37: $P_S^i \leftarrow \text{count}_S^i / M, \quad P_I^i \leftarrow \text{count}_I^i / M, \quad P_R^i \leftarrow \text{count}_R^i / M$
38: **return** $P_S, P_I, P_R$

---

[3]`https://networkx.org/`

## C.3 Details of PID Algorithm

---

**Algorithm 2: PID: Probabilistic Infection Dynamics**

---

**Require:** Graph $\mathcal{G} = (\mathcal{V}, \mathcal{E})$, infection rate $\beta$, recovery rate $\gamma$, initial infected set $\mathcal{I}_0$, threshold $\varepsilon$

**Ensure:** Marginal probabilities $P_S(t), P_I(t), P_R(t)$

1: $P_S^i(0) \leftarrow 1, \ P_I^i(0) \leftarrow 0, \ P_R^i(0) \leftarrow 0, \quad \forall i \in \mathcal{V}$

2: **for** $i \in \mathcal{I}_0$ **do**

3: $\qquad P_S^i(0) \leftarrow 0, \quad P_I^i(0) \leftarrow 1$

4: **end for**

5: $t \leftarrow 0$

6: **repeat**

7: $\qquad$ **for all** $i \in \mathcal{V}$ **do**

8: $\qquad\qquad P_S^i(t+1) \leftarrow P_S^i(t) \cdot \prod_{j \in \mathcal{NE}^{\text{in}}(i)} \left(1 - \beta \cdot P_I^j(t)\right)$

9: $\qquad\qquad P_I^i(t+1) \leftarrow P_S^i(t) - P_S^i(t+1) + (1-\gamma) \cdot P_I^i(t)$

10: $\qquad\qquad P_R^i(t+1) \leftarrow 1 - P_S^i(t+1) - P_I^i(t+1)$

11: $\qquad$ **end for**

12: $\qquad t \leftarrow t+1$

13: **until** $\|P_S(t) - P_S(t-1)\|_2 + \|P_I(t) - P_I(t-1)\|_2 + \|P_R(t) - P_R(t-1)\|_2 \leq \varepsilon$

14: **return** $P_S(t), P_I(t), P_R(t)$

---

## C.4 Details of RAPID Algorithm

---

**Algorithm 3: RAPID: Residual-Accelerated Propagation for Infection Dynamics**

---

**Require:** Graph $\mathcal{G} = (\mathcal{V}, \mathcal{E})$, rates $\beta, \gamma$, initial infected set $\mathcal{I}_0$, threshold $\varepsilon$, preheat steps $p$

**Ensure:** State probabilities $P_S, P_I, P_R$

1: $\qquad\qquad\qquad\qquad\qquad\qquad\qquad\qquad\qquad\qquad\qquad\qquad$ ▷ Initialization

2: $P_S^v \leftarrow 1, \ P_I^v \leftarrow 0, \ P_R^v \leftarrow 0$ for all $v \in \mathcal{V}$

3: **for** $v \in \mathcal{I}_0$ **do**

4: $\qquad P_S^v \leftarrow 0, \quad P_I^v \leftarrow 1$

5: **end for**

6: $\tilde{P}_I^v \leftarrow P_I^v$ for all $v$ $\qquad\qquad\qquad\qquad\qquad\qquad\qquad$ ▷ Snapshot for residuals

7: $\qquad\qquad\qquad\qquad\qquad\qquad$ ▷ Preheat phase to reduce approximation error

8: **for** $t = 1$ to $p$ **do**

9: $\qquad$ **for all** $v \in \mathcal{V}$ **do**

10: $\qquad\qquad \tilde{P}_S^v \leftarrow P_S^v$

11: $\qquad\qquad P_S^v \leftarrow P_S^v \cdot \prod_{u \in \mathcal{NE}^{\text{in}}(v)}(1 - \beta \cdot P_I^u)$

12: $\qquad\qquad P_I^v \leftarrow (1-\gamma) \cdot P_I^v + (\tilde{P}_S^v - P_S^v)$

13: $\qquad\qquad P_R^v \leftarrow 1 - P_S^v - P_I^v$

14: $\qquad$ **end for**

15: **end for**

16: $\qquad\qquad\qquad\qquad\qquad\qquad\qquad\qquad\qquad\qquad$ ▷ Residual initialization

17: **for all** $v \in \mathcal{V}$ **do**

18: $\qquad R_{\text{res}}(v) \leftarrow \beta \cdot \sum_{u \in \mathcal{NE}^{\text{in}}(v)} \left(P_I^u - \tilde{P}_I^u\right)$

19: **end for**

20: Initialize max-heap $\mathcal{H}$ with all $v$ such that $R_{\text{res}}(v) > \varepsilon$

21: **while** $\mathcal{H}$ not empty **do** $\qquad\qquad\qquad\qquad\qquad$ ▷ Residual-driven propagation

22: $\qquad (R_{\text{res}}(i), i) \leftarrow \text{popTop}(\mathcal{H})$

23: $\qquad$ **if** $R_{\text{res}}(i) \leq \varepsilon$ **then break**

24: $\qquad$ **end if**

25: $\qquad$ **if** $P_I^i > \varepsilon$ **then**

26: $\qquad\qquad \tilde{P}_S^i \leftarrow P_S^i$

---

```
27:        $P_S^i \leftarrow P_S^i \cdot \prod_{u \in \mathcal{NE}^{\text{in}}(i)} (1 - \beta \cdot P_I^u)$
28:        $P_I^i \leftarrow (1 - \gamma) \cdot P_I^i + (\tilde{P}_S^i - P_S^i)$
29:        $P_R^i \leftarrow 1 - P_S^i - P_I^i$
30:        $\delta \leftarrow P_I^i - \tilde{P}_I^i$
31:        $\tilde{P}_I^i \leftarrow P_I^i$
32:                                              ▷ Propagate residual to out-neighbors
33:        for $j \in \mathcal{NE}^{\text{out}}(i)$ do
34:            $R_{\text{res}}(j) \leftarrow R_{\text{res}}(j) + \beta \cdot \delta$
35:            if $R_{\text{res}}(j) > \varepsilon$ then push $(R_{\text{res}}(j), j)$ into $\mathcal{H}$
36:            end if
37:        end for
38:      end if
39:      $R_{\text{res}}(i) \leftarrow 0$
40: end while
41: return $P_S, P_I, P_R$
```

## C.5    Design of Propagation Residual

The residual term is derived from the one-step temporal difference of the infected probability, as formulated in Eq. (15):

$$P_I^i(t+1) - P_I^i(t) \lesssim \beta \underbrace{\sum_{j \in \mathcal{V}} \mathbf{A}_{ji}(P_I^j(t) - P_I^j(t-1))}_{\text{Change due to propagation}} + \underbrace{(1 - \gamma)(P_I^i(t) - P_I^i(t-1))}_{\text{Change due to retained infection}}. \qquad (51)$$

Although the propagation residual was originally motivated by the temporal dynamics of infection spread, its formulation is not heuristic. Instead, it admits a principled interpretation as a first-order Jacobian approximation of neighbor-induced perturbations under a linearized infection model.

In the following, we formally derive this connection, show how it naturally leads to the classical epidemic threshold, and provide a physical interpretation of the residual-driven max-heap update mechanism.

**Theorem C.1** (Jacobian Interpretation of Propagation Residual). *In Eq. (16), we define the propagation residual at node $i$ as:*

$$R_{\text{res}}(i) = \beta \sum_{j \in \mathcal{V}} \mathbf{A}_{ji} (P_I^j - \tilde{P}_I^j), \qquad (52)$$

*where $P_I^j$ and $\tilde{P}_I^j$ denote the current and previously cached infection probabilities of node $j$. This residual can be interpreted as a local Jacobian-vector product:*

$$R_{\text{res}}(i) = \sum_{j \in \mathcal{V}} \frac{\partial P_I^i}{\partial P_I^j} \cdot \delta_j, \qquad (53)$$

*where $\delta_j = P_I^j - \tilde{P}_I^j$, and the partial derivative $\frac{\partial P_I^i}{\partial P_I^j} = \beta A_{ji}$ arises from the linear approximation of the infection dynamics. Thus, $R_{\text{res}}(i)$ quantifies the first-order propagation response of node $i$ to perturbations in its neighbors.*

**Remark 4** (Recovery of Epidemic Threshold). *Denote $\Delta P_I(t+1) = P_I^i(t+1) - P_I^i(t)$. Starting from Eq. (C.5), we can express the system in matrix form as:*

$$\Delta P_I(t+1) \lesssim \left(\beta \cdot \mathbf{A}^T + (1 - \gamma) \cdot \mathbf{I}\right) \cdot \Delta P_I(t), \qquad (54)$$

*which defines a linear dynamical system. For this system to remain stable, the spectral radius of the Jacobian matrix must satisfy:*

$$\rho\left(\beta \cdot \mathbf{A}^T + (1 - \gamma) \cdot \mathbf{I}\right) < 1, \qquad (55)$$

*which leads to the classical epidemic threshold condition:*

$$\frac{\beta}{\gamma} < \frac{1}{\lambda_{max}(\mathbf{A})}. \qquad (56)$$

*This derivation illustrates that our residual-based formulation naturally recovers fundamental conditions in epidemic dynamics.*

**Remark 5** (Connection to Local Lipschitz Bound). *The Jacobian matrix $\mathbf{J} = \beta \cdot \mathbf{A}^T$ governs the propagation sensitivity with respect to neighbor infection states. It induces a local Lipschitz upper bound:*

$$\|\Delta P_I(t+1)\| \lesssim \|\mathbf{J}\| \cdot \|\Delta P_I(t)\|. \tag{57}$$

*The residual term $R_{\text{res}}$ captures the Jacobian-induced local propagation energy, and thus provides a structure-aware signal for selective inference.*

**Remark 6** (Connection to Discrete Diffusion Operators). *Although our model targets stochastic epidemic diffusion, the propagation residual exhibits a structural resemblance to classical diffusion operators. Specifically, the residual at node $i$,*

$$R_{\text{res}}(i) = \beta \sum_{j \in \mathcal{V}} \mathbf{A}_{ji} \left( P_I^j - \tilde{P}_I^j \right), \tag{58}$$

*aggregates state changes from incoming neighbors, analogous to the discrete Laplace-Beltrami operator [51]:*

$$\Delta f_i = \frac{1}{d_i} \sum_{j \in \mathcal{N}_i} w_{ij}(f_i - f_j), \tag{59}$$

*which measures local imbalance across a mesh. While not a Laplacian in the strict sense, our formulation shares a key geometric intuition: it emphasizes update prioritization where local dynamics remain active—analogous to high-gradient regions in physical diffusion.*

## D  Experiments

### D.1  Experimental Settings

**Infrastructure and Implementation.** All experiments were conducted on a machine equipped with a Intel(R) Xeon(R) Gold 6248 CPU @ 2.50GHz processor with 376 GB Memory. All algorithms are implemented in Python with the Networkx library.

**Setup of Figure 1.** The setup for each subfigure of Figure 1 in the main paper is as follows: **(a)** We run 20 Monte Carlo simulations on a 1000-node asymmetric Erdős–Rényi graph ($\bar{k} = 10$), varying $\beta$ and $\gamma$ across a $[0.01, 1.0]$ grid. Each point reports the average node-level infection variance, and we plot its relationship with $\log(\beta/\gamma)$ under a fixed initial infection ratio of 10%. **(b)** We simulate disease spread on 1000-node asymmetric Erdős–Rényi graphs with varying target average degree $\bar{k} \in \{0.5, 1, \ldots, 20\}$. For each graph, we randomly infect 10% of nodes and compute the average node-level variance over a $6 \times 6$ grid of $(\beta, \gamma) \in [0.01, 0.2]$. We then analyze the relationship between the actual average degree and the mean variance of the estimator. **(c)** We run 10 Monte Carlo simulations on a 1000-node asymmetric Erdős–Rényi graph ($\bar{k} = 10$), across a $6 \times 6$ grid of $(\beta, \gamma) \in [0.01, 0.2]$. For each initial infection ratio $\alpha \in \{0.01, 0.02, \ldots, 0.30\}$, we compute the mean variance of node-level infection probabilities. **(d)** We run Monte Carlo simulations on a 1000-node asymmetric Erdős–Rényi graph ($\bar{k} = 10$), using an initial infection ratio of 1%. For each sample size $M \in \{5, 10, \ldots, 50\}$, we evaluate the mean estimator variance across a $6 \times 6$ grid of $(\beta, \gamma) \in [0.01, 0.2]$.

**Hyperparameter Setting in Experiments.** For **RAPID**, the number of preheat steps $p$ is set to 20, and the propagation residual threshold is set to $10^{-3}$.

**Full results for Figure 3.** We treat nodes with inferred infection probability above 0.5 as infected. Figure 3 in the main paper shows the classification results on `hiv-Trans` using MC-5, MC-10, PID, and **RAPID**. Here, we present the full results across all six datasets in Figure 5, illustrating each method's ability to distinguish highly infected individuals.

**SIR Parameter Setting of More Diseases.** To ensure the practical relevance of our research, we reviewed the literature and compiled the infectious period and basic reproduction number ($R_0$) data for various infectious diseases, as summarized in Table 5. It is important to note that we assume a well-equipped healthcare environment where infected individuals are isolated and treated promptly upon showing symptoms, effectively preventing further transmission. As a result, for diseases listed

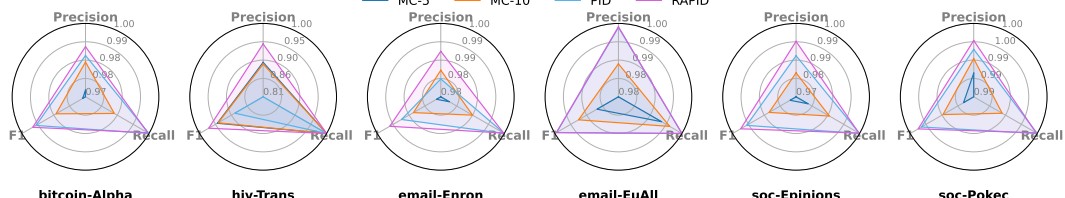

Figure 5: Classification metrics (precision, recall, F1) on six datasets.

in the table other than the common cold, the infectious period is taken to be the incubation period of the disease.

In traditional epidemiological studies, $\gamma$ can also be defined as the rate at which a disease transitions from being infectious to non-infectious. Therefore, $\gamma$ can be calculated as the reciprocal of the infectious period. The SIR model parameters are derived as follows:

$$\gamma = \frac{1}{\text{Infectious Period}}$$
$$\beta = R_0 \cdot \gamma$$

For simplicity, when the infectious period is reported as a range, we use the average of the minimum and maximum values for calculations.

Table 5: Summary of some infectious diseases with their parameters in the SIR model.

| Disease | Infectious Period | Transmission | $R_0$ | $\gamma$ [1/day] | $\beta$ [1/day] |
|---------|-------------------|--------------|-------|------------------|-----------------|
| Nipah Virus | 4–14 days [59] | Body fluids | 0.5 [38] | 0.1111 | 0.0556 |
| Andes Hantavirus | 7–39 days (median 18 days) [55] | Respiratory droplets and body fluids | 1.2 (0.8–1.6) [39] | 0.0556 | 0.0667 |
| MERS | 5 days (2–14 days) [7] | Respiratory droplets | 0.5 (0.3–0.8) [31] | 0.2 | 0.1 |
| Ebola | Average 12.7 days [12] | Body fluids | 1.8 (1.4–1.8) [58] | 0.0787 | 0.1417 |
| Mpox | 4–14 days [59] | Physical contact, body fluids, respiratory droplets, sexual (MSM) | 2.1 (1.1–2.7) [20, 2] | 0.1111 | 0.2333 |

### D.2 Influence of Variance on the Reliability of MC Inference

In this subsection, we aim to bridge the conclusion of Theorem 3.1 with empirical evidence, by demonstrating how a high variance in Monte Carlo (MC) estimators can lead to degraded inference accuracy.

We investigate the relationship between Kendall-$\tau$ and ground-truth per-node infection standard deviation on the `hiv-Trans` dataset by varying the initial infection ratio $\alpha$, as shown in Figure 6. We observe that while MC-based methods exhibit performance degradation as uncertainty increases, RAPID and PID maintain relatively stable ranking performance across high-variance regimes.

### D.3 Extension to Two-stage Non-reinfection Epidemic Model

Although Theorem 3.3 in the main paper shows that our algorithm, while developed under the SIR model, naturally extends to generalized non-reinfection models, in this subsection we provide empirical results on the SEIR model to demonstrate its effectiveness in a more complex setting.

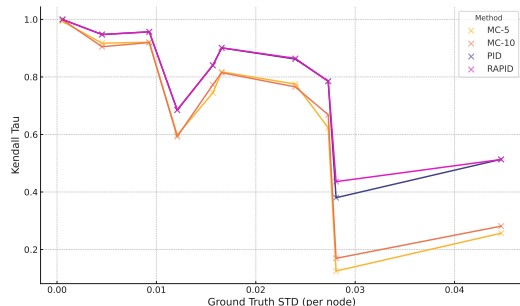

Figure 6: Scatter plot and LOESS-smoothed trend lines of Kendall-$\tau$ versus ground-truth per-node infection standard deviation.

**SEIR Model Dynamics.** The SEIR model extends the classic SIR framework by introducing an intermediate *exposed* (E) state to capture the incubation period of infection. It models a two-stage infection process: individuals first become *exposed* after contact with infectious nodes, and then transition to the *infected* state after a latency period. The continuous-time dynamics are governed by the following ODEs:

$$
\begin{aligned}
\frac{dS}{dt} &= -\beta SI, \\
\frac{dE}{dt} &= \beta SI - \sigma E, \\
\frac{dI}{dt} &= \sigma E - \gamma I, \\
\frac{dR}{dt} &= \gamma I,
\end{aligned}
\tag{60}
$$

where $\beta$ is the infection rate, $\sigma$ is the incubation (exposure-to-infection) rate, and $\gamma$ is the recovery rate. The probability-based update equations are given by:

$$
\begin{aligned}
P_S^i(t+1) &= P_S^i(t) \prod_j (1 - \beta A_{ji} P_I^j(t)), \\
P_E^i(t+1) &= (1-\sigma) P_E^i(t) + P_S^i(t) \left[ 1 - \prod_j (1 - \beta A_{ji} P_I^j(t)) \right], \\
P_I^i(t+1) &= (1-\gamma) P_I^i(t) + \sigma P_E^i(t), \\
P_R^i(t+1) &= P_R^i(t) + \gamma P_I^i(t).
\end{aligned}
\tag{61}
$$

**Extend RAPID to SEIR.** Similar to the idea adopted in the SIR model, for general non-reinfection models, global convergence requires that both $P_E \to 0$ and $P_I \to 0$. In particular, for the two main transition stages $S \to E$ and $E \to I$, we define the following residuals:

- Exposure-stage residual (from $S \to E$):

$$
R_{\text{res}}^E(i) = \beta \sum_{j \in V} A_{ji} \left( P_I^j(t) - P_I^j(t-1) \right).
\tag{62}
$$

- Infection-stage residual (from $E \to I$):

$$
R_{\text{res}}^I(i) = \sigma \left( P_E^i(t) - P_E^i(t-1) \right).
\tag{63}
$$

Note that once $P_E^i \to 0$, the infection-stage residual $R_{\text{res}}^I(i) \to 0$ naturally follows. Moreover, $R_{\text{res}}^I(i)$ depends solely on the change in the node's own state, whereas our main paper defines the propagation residual to be structure-aware. Hence, we define the global propagation residual for non-reinfection models as:

$$
R_{\text{res}}(i) := R_{\text{res}}^E(i) = \beta \sum_{j \in V} A_{ji} \left( P_I^j - \tilde{P}_I^j \right),
\tag{64}
$$

which is exactly the same as our derivation in the main paper for SIR! To verify the effectiveness of our method, we conduct similar experiment in the main paper on `hiv-Trans`.

**Experiment.** We adopt the same parameter setting as in the Nipah Virus scenario, with $\beta = 1/18$ and $\gamma = 1/9$, and further set $\sigma = 0.8$ to model the transition probability from $E \rightarrow I$. The initially infected ratio $\alpha = 0.01$. We use 1000-run Monte Carlo simulation as the ground truth. All other hyperparameter settings are kept consistent with those in the main paper.

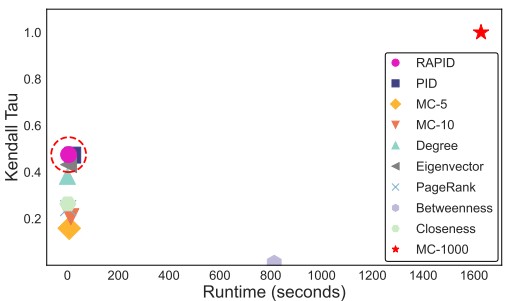

Figure 7: Kendall-$\tau$ vs. Runtime on `hiv-Trans` under SEIR model.

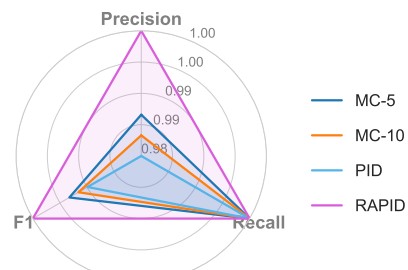

Figure 8: Classification metrics on `hiv-Trans` under SEIR model.

Table 6: SEIR Results on `hiv-Trans` with $\alpha = 0.01$. All values are scaled by $10^{-2}$. Best results are **in bold**.

| Method | hiv-Trans | |
| --- | --- | --- |
| | MAE $\downarrow$ | $t\,(s) \downarrow$ |
| MC-5 | 6.13±0.25 | 3.55±0.21 |
| MC-10 | 4.01±0.35 | 6.79±0.47 |
| PID | 6.23±0.51 | 10.50±0.14 |
| **RAPID** | **0.82**±0.23 | **2.43**±0.12 |

Figure 7 and Figure 8 present the Kendall-$\tau$ versus runtime trade-off and the radar chart of precision, F1 and recall rate under the SEIR model on the `hiv-Trans` dataset, respectively. Table 6 further reports the detailed MAE and runtime values. These results closely mirror those under the SIR setting in the main paper, confirming that our RAPID method remains effective across general non-reinfection models beyond SIR.

### D.4 Influence of Distribution of Initially Infected Nodes

As highlighted in the Introduction of the main paper, traditional epidemic modeling often relies on assumptions of random mixing and population homogeneity, which limit its applicability in real-world scenarios. In contrast, **RAPID** takes the initial infected set $\mathcal{I}_0$ as an explicit input, allowing flexible adaptation to different initial conditions. In this subsection, we demonstrate that **RAPID** remains robust under varying distributions of $\mathcal{I}_0$.

**Impact of Initial Infection Phase $\alpha$.** In this part, we investigate the impact of different values of $\alpha$, which reflect different stages of an epidemic at the initial time. While the main paper presents results for $\alpha = 0.01$ to simulate the early stage which is the most common setting, we additionally evaluate $\alpha = 0.5$ and $\alpha = 0.8$ to represent mid-stage and late-stage outbreaks, respectively, and report the corresponding results.

- $\alpha = 0.5$. The trade-off between Kendall-$\tau$ and runtime is illustrated in Figure 9, while the MAE scores are summarized in Table 7.
- $\alpha = 0.8$. The trade-off between Kendall-$\tau$ and runtime is illustrated in Figure 10, while the MAE scores are summarized in Table 8.

Based on the comparison across different epidemic phases, we make the following observations: (1) Consistent with the results in the main paper, our **RAPID** algorithm maintains high accuracy across

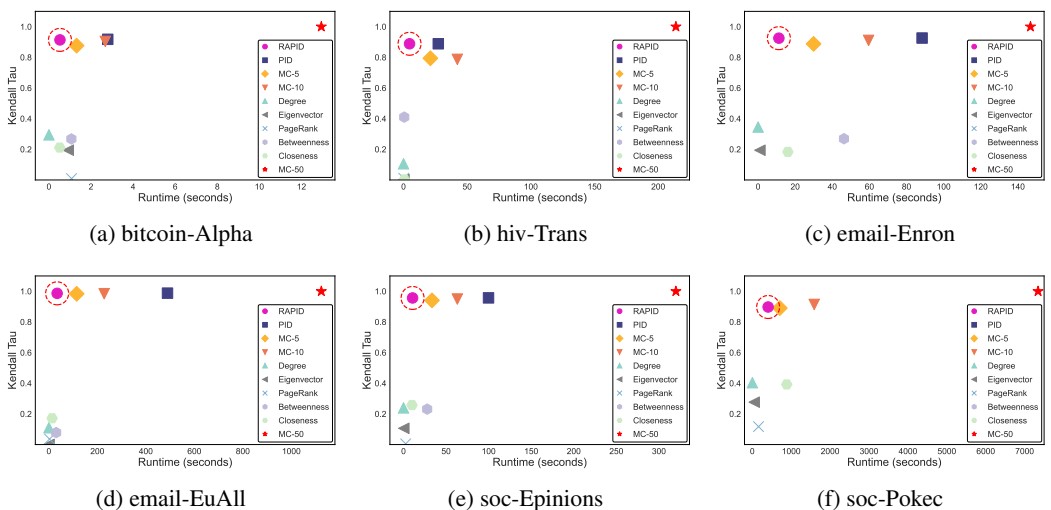

Figure 9: Trade-off between Kendall-Tau and Runtime across six datasets. ($\alpha = 0.5$)

Table 7: MAE comparison ($\alpha = 0.5$, lower is better). All values are scaled by $10^{-2}$. Best results are **in bold**.

| Dataset | MC-5 | MC-10 | PID | **RAPID** |
|---|---|---|---|---|
| bitcoin-Alpha | 4.47±0.02 | 3.39±0.04 | 2.65±0.10 | **2.27**±0.04 |
| hiv-Trans | 4.15±0.11 | 2.14±0.03 | 4.24±0.20 | **1.02**±0.20 |
| email-Enron | 3.99±0.03 | 3.14±0.00 | 2.79±0.01 | **2.26**±0.01 |
| email-EuAll | 1.33±0.01 | 1.02±0.00 | 0.73±0.00 | **0.68**±0.00 |
| soc-Epinions | 2.99±0.02 | 2.36±0.01 | 1.91±0.01 | **1.57**±0.01 |
| soc-Pokec | 2.24±0.00 | 1.76±0.00 | 1.35±0.00 | **1.16**±0.00 |

Figure 10: Trade-off between Kendall-Tau and Runtime across six datasets. ($\alpha = 0.8$)

all $\alpha$ settings, consistently accelerating both MC-based methods and PID, while preserving a runtime comparable to a single-run MC simulation. (2) As the initial infection ratio increases, centrality-based baselines become increasingly unreliable, and by $\alpha = 0.8$, most of them exhibit almost no correlation with ground truth.

**Impact of Initial Infection Clustering Patterns.** In this part, we divide the initial infection set into several disjoint and disconnected clusters to examine the impact on different methods. We conduct experiments on the bitcoin-Alpha dataset with an initial infection ratio of $\alpha = 0.01$. Figure 11 demonstrates the robustness of **RAPID** under varying distributions of initial infection clusters.

Table 8: MAE comparison ($\alpha = 0.8$, lower is better). All values are scaled by $10^{-2}$. Best results are **in bold**.

| Dataset | MC-5 | MC-10 | PID | RAPID |
|---|---|---|---|---|
| bitcoin-Alpha | $1.84_{\pm0.03}$ | $1.34_{\pm0.01}$ | $0.96_{\pm0.02}$ | $\mathbf{0.94}_{\pm0.01}$ |
| hiv-Trans | $1.68_{\pm0.03}$ | $1.26_{\pm0.00}$ | $\mathbf{0.31}_{\pm0.05}$ | $0.45_{\pm0.07}$ |
| email-Enron | $1.62_{\pm0.02}$ | $1.29_{\pm0.01}$ | $0.92_{\pm0.01}$ | $\mathbf{0.87}_{\pm0.01}$ |
| email-EuAll | $0.57_{\pm0.00}$ | $0.45_{\pm0.00}$ | $0.30_{\pm0.00}$ | $\mathbf{0.29}_{\pm0.00}$ |
| soc-Epinions | $1.21_{\pm0.00}$ | $0.95_{\pm0.00}$ | $0.68_{\pm0.01}$ | $\mathbf{0.63}_{\pm0.01}$ |
| soc-Pokec | $0.89_{\pm0.00}$ | $0.70_{\pm0.00}$ | $0.49_{\pm0.00}$ | $\mathbf{0.47}_{\pm0.00}$ |

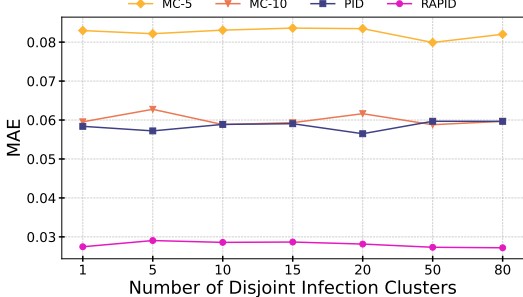

Figure 11: Expected MAE vs. Number of Disjoint Infection Clusters.

### D.5 Results for more Diseases

According to Remark 2, since the discrete SIR simulation aims to approximate a continuous-time process, we require $\beta \ll 1$ and $\gamma \ll 1$. Under this regime, the simulation behavior depends primarily on the ratio $R_0 = \beta/\gamma$, rather than the specific values of $\beta$ and $\gamma$, which aligns with classical epidemiological theory. Therefore, in our additional experiments, we evaluate the algorithm's performance using representative values of $R_0$ listed in Table 5.

**Andes Hantavirus** ($R_0 = 1.2$). The Kendall-$\tau$ versus runtime trade-off is shown in Figure 12, and the MAE comparison across different datasets is presented in Figure 9.

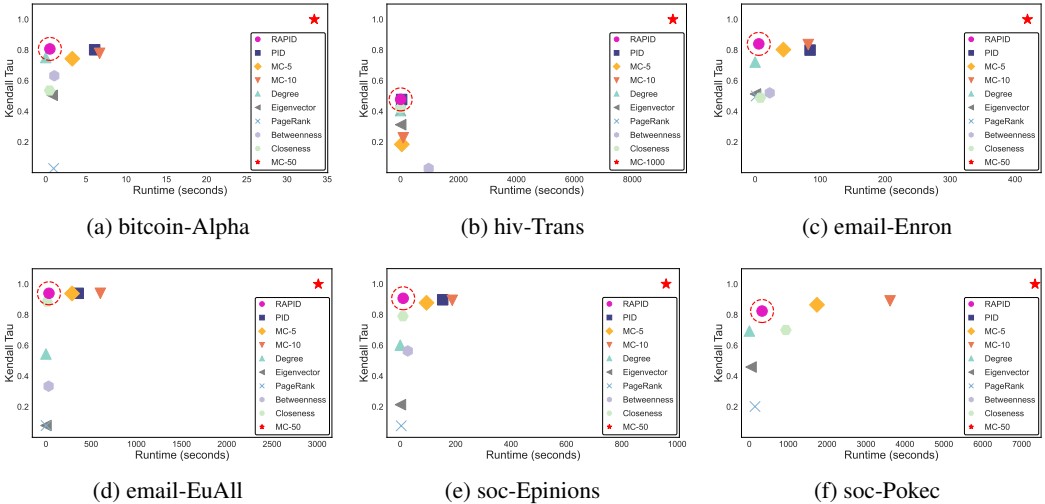

Figure 12: Trade-off between Kendall-Tau and Runtime across six datasets. **(Andes Hantavirus)**

**MERS** ($R_0 = 0.5$). As MERS share the same $R_0$ with Nipah Virus according to Table 5, the results are the same as those in main paper.

**Ebola** ($R_0 = 1.8$). The Kendall-$\tau$ versus runtime trade-off is shown in Figure 13, and the MAE comparison across different datasets is presented in Figure 10.

Table 9: MAE comparison (**Andes Hantavirus**, lower is better). All values are scaled by $10^{-2}$. Best results are **in bold**.

| Dataset | MC-5 | MC-10 | PID | **RAPID** |
|---|---|---|---|---|
| bitcoin-Alpha | $8.21_{\pm0.10}$ | $6.53_{\pm0.03}$ | $8.27_{\pm0.06}$ | $\mathbf{6.38}_{\pm0.05}$ |
| hiv-Trans | $14.82_{\pm0.30}$ | $\mathbf{4.99}_{\pm0.12}$ | $16.79_{\pm0.35}$ | $5.40_{\pm0.36}$ |
| email-Enron | $6.82_{\pm0.08}$ | $5.26_{\pm0.01}$ | $9.31_{\pm0.02}$ | $\mathbf{4.52}_{\pm0.08}$ |
| email-EuAll | $2.13_{\pm0.01}$ | $1.69_{\pm0.02}$ | $2.05_{\pm0.03}$ | $\mathbf{1.17}_{\pm0.02}$ |
| soc-Epinions | $5.47_{\pm0.03}$ | $4.33_{\pm0.05}$ | $6.96_{\pm0.01}$ | $\mathbf{4.31}_{\pm0.03}$ |
| soc-Pokec | $3.64_{\pm0.00}$ | $\mathbf{2.86}_{\pm0.00}$ | $3.68_{\pm0.01}$ | $3.51_{\pm0.01}$ |

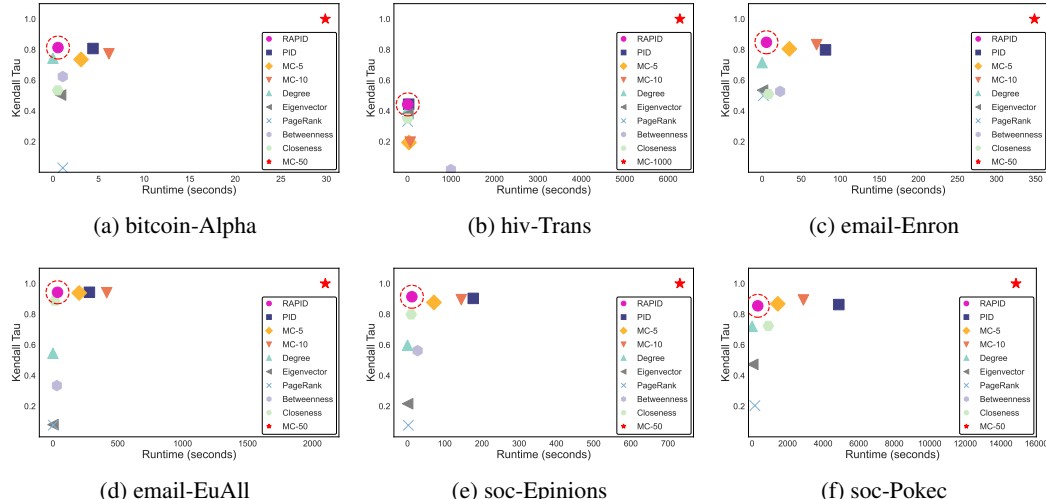

(a) bitcoin-Alpha     (b) hiv-Trans     (c) email-Enron

(d) email-EuAll     (e) soc-Epinions     (f) soc-Pokec

Figure 13: Trade-off between Kendall-Tau and Runtime across six datasets. **(Ebola)**

Table 10: MAE comparison (**Ebola**, lower is better). All values are scaled by $10^{-2}$. Best results are **in bold**.

| Dataset | MC-5 | MC-10 | PID | **RAPID** |
|---|---|---|---|---|
| bitcoin-Alpha | $8.79_{\pm0.09}$ | $6.82_{\pm0.08}$ | $7.10_{\pm0.11}$ | $\mathbf{4.77}_{\pm0.08}$ |
| hiv-Trans | $10.50_{\pm0.44}$ | $6.29_{\pm0.23}$ | $3.31_{\pm0.72}$ | $\mathbf{1.87}_{\pm0.62}$ |
| email-Enron | $7.49_{\pm0.04}$ | $5.98_{\pm0.01}$ | $9.27_{\pm0.04}$ | $\mathbf{4.39}_{\pm0.01}$ |
| email-EuAll | $2.10_{\pm0.00}$ | $1.67_{\pm0.01}$ | $1.46_{\pm0.01}$ | $\mathbf{1.02}_{\pm0.01}$ |
| soc-Epinions | $5.51_{\pm0.02}$ | $4.30_{\pm0.01}$ | $5.41_{\pm0.02}$ | $\mathbf{2.99}_{\pm0.01}$ |
| soc-Pokec | $4.33_{\pm0.00}$ | $3.40_{\pm0.00}$ | $3.42_{\pm0.00}$ | $\mathbf{2.68}_{\pm0.00}$ |

**Mpox** ($R_0 = 2.1$). The Kendall-$\tau$ versus runtime trade-off is shown in Figure 14, and the MAE comparison across different datasets is presented in Figure 11.

Table 11: MAE comparison (**Mpox**, lower is better). All values are scaled by $10^{-2}$. Best results are **in bold**.

| Dataset | MC-5 | MC-10 | PID | **RAPID** |
|---|---|---|---|---|
| bitcoin-Alpha | $5.44_{\pm0.03}$ | $4.20_{\pm0.07}$ | $2.71_{\pm0.02}$ | $\mathbf{2.68}_{\pm0.01}$ |
| hiv-Trans | $7.44_{\pm0.03}$ | $6.03_{\pm0.04}$ | $1.84_{\pm0.61}$ | $\mathbf{1.78}_{\pm0.66}$ |
| email-Enron | $5.57_{\pm0.01}$ | $4.37_{\pm0.02}$ | $2.83_{\pm0.01}$ | $\mathbf{2.77}_{\pm0.01}$ |
| email-EuAll | $0.96_{\pm0.00}$ | $0.76_{\pm0.00}$ | $0.48_{\pm0.00}$ | $\mathbf{0.47}_{\pm0.00}$ |
| soc-Epinions | $2.84_{\pm0.01}$ | $2.24_{\pm0.00}$ | $1.47_{\pm0.01}$ | $\mathbf{1.45}_{\pm0.01}$ |
| soc-Pokec | $6.74_{\pm0.00}$ | $5.29_{\pm0.00}$ | $3.58_{\pm0.00}$ | $\mathbf{3.52}_{\pm0.00}$ |

**Discussion.** (1) Across all epidemic settings and datasets, our RAPID method consistently maintains high Kendall-$\tau$ and low MAE, often achieving the best performance while significantly improving computational efficiency. (2) Under the Andes Hantavirus setting ($R_0 = 1.2$), we observe notably higher variance on the `hiv-Trans` dataset, aligning with Theorem 3.1 in the main paper. However, this increased stochasticity also challenges the fidelity of our reference distribution. In our experi-

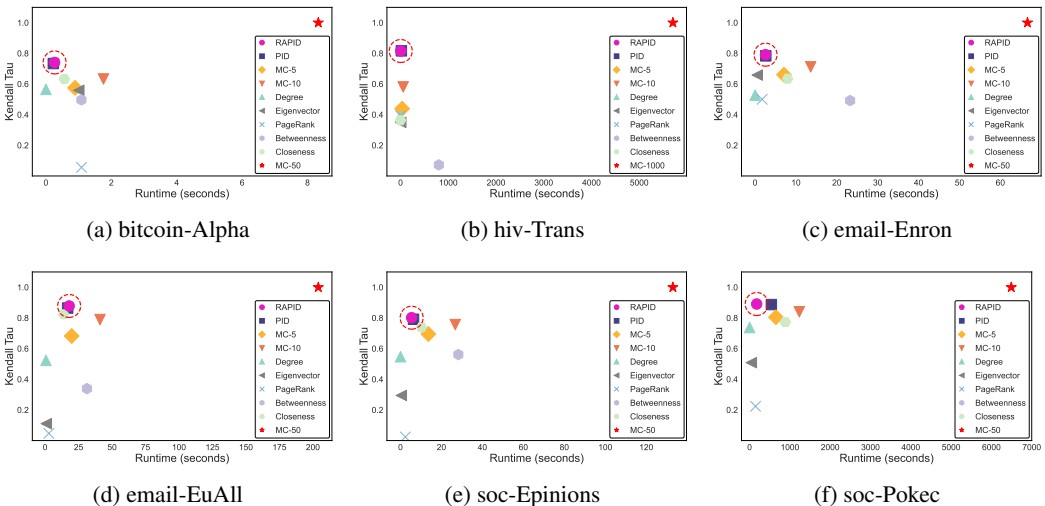

Figure 14: Trade-off between Kendall-Tau and Runtime across six datasets. **(Mpox)**

ments, we estimate the ground truth infection probability $\bar{P}_{\text{GT}}$ using 50 Monte Carlo simulations (on `hiv-Trans`, we use 1000-run Monte Carlo simulation as ground-truth). While this yields accurate approximations on most datasets, the elevated randomness in `hiv-Trans` makes the reference distribution inherently noisy. As a result, the MAE values should be interpreted with caution, as they may partially reflect the variance of the reference rather than the bias of the estimators.

### D.6 Sensitivity of Hyperparameters

**RAPID** has two key hyperparameters: the number of preheat steps $p$ used to initialize the inference via PID, and the propagation residual threshold $\varepsilon$ used to control message updates. Figure 15 shows how these hyperparameters affect performance in terms of mean Kendall-$\tau$ and MAE.

We observe that increasing $p$ initially improves both Kendall-$\tau$ and MAE, as more preheating facilitates better initialization. However, beyond a moderate value, the performance begins to degrade, indicating that excessive preheating may lead to over-propagation and reduced inference quality. For the residual threshold $\varepsilon$, we observe that overly loose settings (i.e., large $\varepsilon$) lead to reduced performance, as many small but informative updates are skipped. However, as $\varepsilon$ becomes very small, its influence diminishes. This is because the scale of local residuals is largely determined by $\bar{k}\beta$, so further reducing $\varepsilon$ without changing $\beta$ has limited effect. Overall, **RAPID** maintains stable performance across a wide range of settings, demonstrating strong robustness to both hyperparameters.

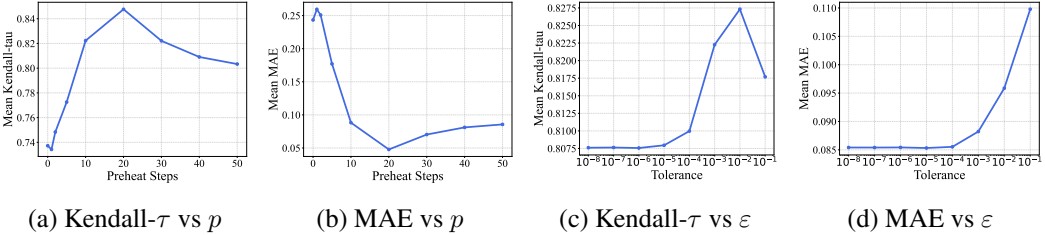

Figure 15: Sensitivity analysis on preheat steps $p$ and propagation residual threshold $\varepsilon$.

## E More Discussion

### E.1 Limitation

Although **RAPID** achieves high inference accuracy and significant runtime reduction across various networks, several limitations remain. First, our framework assumes a discrete-time approximation of the continuous-time infection process, and the linearization results rely on sufficiently small

per-time-step transmission and recovery probabilities, i.e., $\beta, \gamma \ll 1$ (see Remark 2). In practical scenarios with coarse time resolution or unscaled parameters, this assumption may not strictly hold, potentially introducing approximation bias. Second, **RAPID** is designed under the assumption of non-reinfection epidemic models (e.g., SIR, SEIR), where infections are irreversible. Extending the method to reinfection settings such as SIS or SIRS models remains non-trivial, as it would require fundamentally different update dynamics and potentially new convergence guarantees.

## E.2    Future Work

**Extension to General Epidemic Settings.** While our method is developed under the setting of non-reinfection models, extending it to more general epidemic settings is a promising direction. For reinfective models (e.g., SIS or SIRS), where nodes can return to the susceptible state, the propagation process becomes recurrent and requires modifying the residual formulation to account for cyclical transitions. Moreover, real-world epidemics often involve time-varying parameters—such as dynamic infection or recovery rates $\beta(t)$ and $\gamma(t)$—which can be incorporated into our framework by adapting the update rules to operate on temporal sequences of residuals. Finally, in dynamic graphs where nodes and edges evolve over time (e.g., contact networks or mobility patterns), our residual-driven approach can be extended by computing propagation residuals on each snapshot and reinitializing the active node queue accordingly. Together, these extensions would enable our method to handle a wider range of realistic and complex epidemic processes.

**Extension to Graph Neural Networks (GNNs).** Our theoretical framework of linear approximation, residual-driven selection, and asynchronous propagation can be naturally extended to enhance Graph Neural Networks. Specifically, message passing in GNNs often involves nonlinear transformations, which can similarly benefit from Jacobian-based linear approximations (explained in Theorem C.1) to quantify node-level sensitivity to neighbor perturbations. By defining propagation residuals based on node feature dynamics, one can design prioritized, residual-guided asynchronous GNNs that selectively update nodes according to their local propagation significance. Such an approach promises significant computational efficiency improvements on large-scale graphs and offers theoretical insights into the interpretability and stability of GNNs. Exploring this connection presents an exciting avenue for future research.

