# OpenReview forum: "Scaling Epidemic Inference on Contact Networks: Theory and Algorithms"
_NeurIPS.cc/2025/Conference — NeurIPS 2025 poster_

### Official Review · Reviewer_wj6a · 2025-06-24

**Clarity:** 3
**Significance:** 1
**Originality:** 1
**Rating:** 2
**Confidence:** 4

**Summary:**

This paper addresses the problem of estimating node-level infection probabilities at convergence under the stochastic SIR (Susceptible-Infected-Recovered) model on large contact networks. It begins by deriving a lower bound on the variance of Monte Carlo (MC) estimators, highlighting the impact of transmission parameters and network structure on simulation cost. The authors then propose a linear approximation to the nonlinear infection dynamics, and prove a convergence condition for non-reinfection models. Building on these theoretical insights, the paper introduces RAPID, a deterministic and residual-aware propagation algorithm that selectively updates nodes based on estimated influence from neighbors. RAPID is empirically evaluated on six real-world networks, showing comparable accuracy to multi-run MC simulations with significantly reduced runtime.

**Questions:**

1. **Clarify relevance to NeurIPS.** The paper currently reads as a computational epidemiology or network modeling study, with no learning component and limited connection to the broader ML community. If there is a machine learning angle—e.g., inference under uncertainty, approximation of stochastic processes, or use of RAPID as part of a learning pipeline—please make this connection explicit. Without it, it's difficult to see how the work fits within the NeurIPS scope.

2. **Justify novelty beyond existing approximation methods.** The use of linear approximations and deterministic propagation resembles techniques already present in the literature (e.g., Fokker–Planck, mean-field approximations, and message passing). Please clarify what’s truly new here: either in the theoretical analysis, the algorithmic contribution, or the application framing.

3. **Use more realistic contact network data.** The current experiments are based on communication or social graphs that don’t reflect how physical contact or transmission actually occurs. To better support the practical value of RAPID, please consider evaluating on more appropriate datasets—e.g., **time-resolved proximity networks** from **Sociopatterns** (e.g., high school, hospital, workplace settings), **mobility-derived contact graphs** from datasets like **SafeGraph**, or **Bluetooth-based contact traces** (e.g., from MIT's Reality Mining or COVID-related studies). Showing that RAPID produces meaningful predictions on these real-world contact networks—despite soundness concerns—could help demonstrate practical utility.

4. **Clarify when your approximations are accurate.** The theoretical results rely on assumptions like independence of infection paths and small infection probabilities. Can you provide intuition or empirical evidence on when your approximations hold or break down?

**What could change my evaluation:** A clear and convincing explanation of how this work contributes to machine learning, along with stronger justification for its novelty and practical impact, would help address my main concerns.

**Ethical Concerns:**

["NO or VERY MINOR ethics concerns only"]

**Final Justification:**

This paper presents RAPID, a deterministic algorithm for scalable epidemic inference on contact networks. The method is technically sound, well implemented, and clearly explained. I appreciate the authors’ responsiveness during the discussion and their efforts to improve the empirical scope, including the addition of a real-world dataset and clarification of theoretical assumptions.

That said, my evaluation remains unchanged. The paper does not involve learning, estimation, or data-driven modeling, and its connection to machine learning—as typically represented at NeurIPS—is limited. While the algorithm may be valuable within epidemic modeling or network science, I do not find the contribution sufficiently aligned or impactful in the context of modern ML research. I therefore maintain my initial rating.

**Limitations:**

No. The paper does not discuss potential negative societal impact, though it is unlikely to have one given the focus on epidemic simulation.

**Paper Formatting Concerns:**

I haven't noticed any formatting concerns.

**Quality:**

2

**Strengths And Weaknesses:**

**Strengths:** The paper is clearly written, well-organized, and technically competent. The authors thoughtfully implement a deterministic approximation algorithm (RAPID) for node-level inference in SIR epidemic models and benchmark it on several real-world networks. The residual-based propagation idea is a reasonable engineering technique to improve computational efficiency, and the empirical runtime comparisons are thorough. The paper also provides a self-contained narrative connecting theoretical motivation, algorithm design, and experiments, which helps with clarity.

**Weaknesses:** Despite its technical completeness, the paper lacks significance and relevance to the NeurIPS community. It is not a machine learning paper in any meaningful sense—there is no learning component, no connection to generative modeling, deep learning, or agentic AI—and it is difficult to imagine which NeurIPS area the paper would fit into if accepted. The core problem of estimating infection probabilities under SIR dynamics is well-studied, and the approximations proposed here (e.g., linearization of nonlinear dynamics, residual propagation) are reminiscent of existing approaches such as the Fokker–Planck formulation of Langevin dynamics and PDE-based moment matching methods. The theoretical results rely heavily on strong assumptions (e.g., independence between infection paths, mean-field approximations), which cast doubt on the validity of the variance bound and the linearized update at the node level. These assumptions are neither controlled nor justified, and the results risk being misleading if interpreted as rigorous. The convergence theorem is also superficial, essentially restating a trivial absorbing property of SIR processes. On the empirical side, the datasets used are either outdated or loosely related to actual contact networks, and more realistic options (e.g., temporal mobility or sensor-based proximity networks) are overlooked. Overall, the paper offers neither new theoretical insight nor practical modeling contributions that rise to the level of significance expected at NeurIPS.

---

> ### Author Rebuttal · Authors · 2025-07-30
>
> We sincerely appreciate your efforts to review our paper and provide valuable suggestions. Below we address each concern in detail.
>
> ---
> - q1: Clarify relevance to NeurIPS.
> - **R1**: Thank you for the comment regarding NeurIPS relevance. We respectfully point out that the NeurIPS 2025 Call for Papers explicitly welcomes research in probabilistic methods, scalable inference, optimization, theory, and machine learning for sciences—including health, network science, and the natural sciences. Our work addresses a core challenge in efficient, deterministic inference for structured infection stochastic processes on large networks, bridging algorithmic theory, scalable probabilistic modeling, and ML for science. We believe RAPID addresses a key bottleneck for **scalable probabilistic inference under uncertainty on networks**, which are central to many areas of modern ML.
>
> Furthermore, probabilistic inference on graphs is a classic topic in machine learning as evidenced by Belief Propagation (BP), which was first published at AAAI’82 [1] and remains foundational in the field. We thus believe that our RAPID algorithm and its advances in scalable, deterministic inference will be of significant interest to the NeurIPS community.
>
> ---
> - q2: Justify novelty beyond existing approximation methods.
> - **R2**: As stated in our introduction, our contribution is twofold: (1) we derive a **variance lower bound** for Monte Carlo estimators on large contact networks, revealing how reproduction number, average out-degree, initial infection ratio, and simulation count affect estimator variance. This result is also empirically validated in Figure 1; (2) we develop **RAPID**, an efficient epidemic inference algorithm that innovatively replaces the residual in classical LocalPush algorithms with Jacobian-based perturbation terms, enabling **asynchronous, selective updates** instead of synchronous iterations. This residual-driven approach achieves **4-12x** computational speedup while maintaining accuracy. We also establish worst-case complexity analysis and derive connections between propagation residuals and epidemic thresholds. Extensive experiments validate our method's effectiveness and efficiency. Reviewers iUqy and MyzN both recognized the novelty and practical impact of our algorithm.
>
> Regarding the approximation methods mentioned, we have clarified their relationship to our algorithm in the preliminaries and related work. **Belief Propagation** (a variant of message passing) operates on factor graphs with sum-product updates for general probabilistic inference; in fact, our PID baseline (Algorithm 1) is equivalent to the BP implementation for epidemic diffusion on contact networks, as in [6]. **Mean-field approximation** assumes independence among variables to simplify joint distributions and forms the foundation of traditional MP/BP methods. **Fokker–Planck equations** describe the global density evolution of continuous systems via partial differential equations, and are unrelated to our discrete, network-based approach.
>
> RAPID advances beyond epidemic BP by **pioneering the use of a local push framework and Jacobian-based residuals for forward inference in nonlinear dynamics**. This asynchronous, selective propagation strategy is not only novel but also highly efficient, making RAPID widely applicable to disease modeling, rumor spreading, and general network-based diffusion processes that require scalable, deterministic inference.
>
> ---
> - q3: Use more realistic contact network data.
> - **R3**: Thank you for the suggestion. Contact networks derived from mobility data providers like SafeGraph would indeed be valuable. However, we have also considered realistic settings by using hiv-Trans, a real-world HIV transmission network from the U.S. (1988-2001) collected by the National Addiction & HIV Data Archive Program [2]. Other datasets like email-EuAll and bitcoin-Alpha are commonly used for epidemic simulations [3,4], as viral spread shares similarities with rumor propagation and computer viruses. Following your suggestion, we added a hospital contact network derived from patient-provider interactions at Carilion Hospital, Virginia [5]. This dataset contains 11,810 nodes and 30,994 edges (undirected edges converted to bidirectional). Results compared against MC-1000 (for sufficiently small variance) are shown below, and we will incorporate these findings into Figure 2, Figure 4, and Tables 2-4.
>
> |             | Kendall-tau ($\times 10^{-2}$) | MAE ($\times 10^{-2}$) | Runtime (s)     |
> | ----------- | ------------------------------ | ---------------------- | --------------- |
> | **RAPID**   | **56.77±0.01**                 | **2.64±0.49**          | **1.07 ± 0.00** |
> | PID         | 56.76±0.01                     | 5.67±0.51              | 3.56 ± 0.01     |
> | MC-5        | 30.80±0.83                     | 13.01±0.35             | 5.81±0.51       |
> | MC-10       | 32.73±0.06                     | 10.78±0.44             | 13.46±1.14      |
> | Degree      | 54.28±0.01                     | --                     | 0.00± 0.00      |
> | Eigenvector | 29.78±0.00                     | --                     | 0.33± 0.00      |
> | PageRank    | 54.28±0.01                     | --                     | 0.18± 0.01      |
> | Betweenness | 47.80±0.01                     | --                     | 2.77± 0.02      |
> | Closeness   | 29.18±0.06                     | --                     | 1.10± 0.01      |
>
> ---
> - q4: Clarify when your approximations are accurate.
> - **R4**: Our approximations are accurate when the contact network is **sparse or locally tree-like**. As mentioned in Remark 1 (Appendix), setting epidemic parameters $\beta$ and $\gamma$ to sufficiently small and proportionally reduced values ensures the linear approximation is accurate when modeling continuous infection processes. The primary source of error comes from the independent path assumption in Algorithm 1, which introduces inaccuracies due to non-independent path interactions (such as network loops). However, we emphasize that this is a standard and widely adopted assumption in network-based epidemic modeling.  [6] shows Algorithm 1 is exact on tree graphs and provides rigorous upper bounds for infected individuals on arbitrary networks. [7] proves it typically yields accurate estimations on real sparse networks for a broad class of spreading dynamics. Therefore, RAPID's approximation errors are primarily related to graph sparsity and local structure: it achieves high accuracy on commonly encountered sparse networks or locally tree-like graphs, while on denser networks it may overestimate marginal infection probabilities—but still provides a meaningful worst-case estimate, which is valuable for robust epidemic assessment. As shown in Table 3, RAPID achieves **highly accurate estimates with MAE of 0.0127 and 0.0103 on the sparsest networks** hiv-Trans and email-EuAll, respectively.
>
>
> ---
> [1] Pearl, Judea. "Reverend Bayes on inference engines: A distributed hierarchical approach." Proceedings of the Second AAAI Conference on Artificial Intelligence (AAAI-82), 1982, pp. 133–136.
>
> [2] Morris, Martina, and Richard Rothenberg. "Hiv transmission network metastudy project: An archive of data from eight network studies, 1988--2001." (2011).
>
> [3] He, Yinhan, et al. "Demystify Epidemic Containment in Directed Networks: Theory and Algorithms." Proceedings of the Eighteenth ACM International Conference on Web Search and Data Mining. 2025.
>
> [4] Ru, Xiaolei, et al. "Inferring patient zero on temporal networks via graph neural networks." Proceedings of the AAAI Conference on Artificial Intelligence. Vol. 37. No. 8. 2023.
>
> [5] Adhikari, Bijaya, et al. "Fast and near-optimal monitoring for healthcare acquired infection outbreaks." PLoS computational biology 15.9 (2019): e1007284.
>
> [6] Karrer, Brian, and Mark EJ Newman. "Message passing approach for general epidemic models." Physical Review E—Statistical, Nonlinear, and Soft Matter Physics 82.1 (2010): 016101.
>
> [7] Lokhov, Andrey Y., Marc Mézard, and Lenka Zdeborová. "Dynamic message-passing equations for models with unidirectional dynamics." Physical Review E 91.1 (2015): 012811.

---

> > ### Comment · Reviewer_wj6a · 2025-08-04
> >
> > Thank you to the authors for the thoughtful and detailed rebuttal. I appreciate your responsiveness and constructive engagement throughout the discussion. While I remain unconvinced that the work aligns with core machine learning contributions as typically represented at NeurIPS, and find the significance somewhat limited in that context, I recognize the technical soundness of the contribution and the clarity with which it is presented. I can see how the work may be viewed as a valuable contribution within its intended scope.

---

### Official Review · Reviewer_7Sx7 · 2025-07-01

**Clarity:** 3
**Significance:** 3
**Originality:** 3
**Rating:** 5
**Confidence:** 4

**Summary:**

This paper addresses a stochastic graph/contact network-based epidemic inference problem with the SIR equations as the backbone. The problem is coined \textbf{"Individual-Level Epidemic Inference"}, where the goal is to estimate the steady-state probability distribution of the 3 states: Susceptible, Infectious and Recovered for each node in the graph.

In particular, the authors first analyze the efficiency (variance of the estimator) of Monte Carlo-based simulation methods with respect to the network structure and SIR model parameters. Then, they analyze a proposed baseline Propagation for Infection Dynamics (\textbf{PID}) that relies on message passing updates of the SIR probabilities over time and show the neighborhood-level approximations and global convergence, which are then utilized to derive a more computationally efficient algorithm coined Residual-Aware Propagation for Infection Dynamics (\textbf{RAPID}). Relying on the theoretical results of convergence, they define a \textbf{propagation residual} term that is used to determine whether to update a node or not, thereby lowering the overall number of nodes that need to be updated. Moreover, worst-case time complexity is derived for RAPID.

Empirical results include a variety of graph-based datasets, including an undirected graph, and is based on the SIR parameters of the Nipah Virus. The accuracy and computational efficiency of RAPID are demonstrated and are clearly more superior than both Monte Carlo simulation and PID.

**Questions:**

- Although it is not explained in the paper (which it should be), the Monte Carlo simulations have stochasticity coming from random sampling of the initial infection node set (as I found out in the code https://anonymous.4open.science/r/rapid-8080/simulations/mc.py). What is the reason for not doing this for PID or RAPID? I would have thought that different initial conditions may yield different steady state results, and therefore you can get a distribution over the 3 states over all the random seeds?

- Can this framework be transferred into the Bayesian inference setting with infection/deaths observations of pathogens (e.g. COVID-19)?

- Why is RAPID's accuracy higher than PID, considering that it is an approximation to PID? Is it because it is converging faster than PID? But then shouldn't all models be run until convergence in your experiments?

- It is said that the ground-truth infection probabilities are calculated using 50-run Monte Carlo simulations. How accurate are these results (maybe hard to tell as this is a simulation) and how big are the variance? If the variance is not negligible, then it is not a very reliable ground-truth.

**Ethical Concerns:**

["NO or VERY MINOR ethics concerns only"]

**Final Justification:**

I am satisfied with the authors response to my concerns and there is sufficient additional evidence to improve my score

**Limitations:**

- I think there is not enough discussion of the limitations of both their modeling framework and the proposed RAPID algorithm, particular its application to real world problems such as disease modeling.
- The variance is mentioned at the beginning for Monte Carlo, but then not analyzed for PID and RAPID.
- I see no issues with negative societal impacts of the work since this is pure methodological work

**Paper Formatting Concerns:**

- Line 113 has a repeated "variance of"
- Line 76 has a repeated "state"
- Table 2: Dataset statistics

**Quality:**

3

**Strengths And Weaknesses:**

Strengths:
- Many theorems are proposed to address the properties of Monte Carlo simulation and PID

- The efficiency and accuracy are clear compared to the selected baselines

- The proposed algorithms are neatly implemented/proposed and they are easy to follow and justify (based on the theoretical results)

Weaknesses:
- The variance lower bound is proved in Theorem 3.1 for Monte Carlo, but it is not clear what purpose it serves as there is no comparison to PID or RAPID, both empirically or theoretically. Empirically, it would at least help if the confidence intervals were shown.

- The experiments are all somewhat "synthetic" since they use the Nipah Virus parameters on different graphs. Therefore I think there needs to be better and more realistic experimental settings that make the case that RAPID can be used for infectious disease modeling. In addition, the ground-truth is taken to be the Monte Carlo simulation results, but it would have been better if the results were compared against actual observations (e.g. if modeling COVID spread [1], you can look at the actual observed infection/death counts, $R_t$ values or just the general trend of infections).

- I also think the vast literature of inference of SIR models using Bayesian inference (e.g. approximate Bayesian computation (ABC) or MCMC) are not mentioned or compared with at all, reducing the paper's significance.

[1] https://www.imperial.ac.uk/media/imperial-college/medicine/sph/ide/gida-fellowships/Imperial-College-COVID19-NPI-modelling-16-03-2020.pdf

---

> ### Author Rebuttal · Authors · 2025-07-30
>
> We sincerely appreciate your efforts to review our paper and provide valuable suggestions. Below we address each concern in detail.
>
> ---
> - w1: Purpose for the variance lower bound proved in Theorem 3.1 for Monte Carlo.
> - **R w1**: The variance lower bound proved in Theorem 3.1 for Monte Carlo both motivates the development of efficient deterministic methods and highlights scenarios where MC simulations become computationally expensive—precisely the cases where RAPID offers the greatest advantage. Theorem 3.1 analyzes how Monte Carlo estimator variance depends on network structure and epidemic parameters $\beta,\gamma$, showing that under certain conditions, this variance can become very large. This high variance is intrinsic to the network itself, as discussed in Remark 1 and empirically validated on ER graphs in Figure 1. For example, small-world networks like hiv-Trans (sparse with small diameter) show higher MC simulation variance than other networks like email-EuAll under identical settings (also shown in **R4**), necessitating MC-1000 for stable, reliable results.
>
> In contrast, PID and RAPID are deterministic methods that do not involve sampling, so they do not have estimator variance or confidence intervals. Their accuracy depends only on approximation error with respect to the true infection probabilities, not on stochastic variation.
>
> ---
> - w2: The experiments are all somewhat "synthetic".
> - **R w2**: Due to its theoretical nature, both classical [5,6] and modern work [2,3] in network-based epidemiology rely heavily on simulation-based validation. However, we have considered **realistic epidemic settings** by using hiv-Trans, a real-world HIV transmission network from the U.S. (1988-2001) collected by the National Addiction & HIV Data Archive Program [1]. Other datasets like email-EuAll and bitcoin-Alpha are commonly used for epidemic simulations [2,3], as viral spread shares similarities with rumor propagation and computer viruses propagation. Following your suggestion, we added a hospital contact network derived from patient-provider interactions at Carilion Hospital, Virginia [4]. This dataset contains 11,810 nodes and 30,994 edges (undirected edges converted to bidirectional). Results compared against MC-1000 (for sufficiently small variance) are shown below, and we will incorporate these findings into Figure 2, Figure 4, and Tables 2-4.
>
> |             | Kendall-tau ($\times 10^{-2}$) | MAE ($\times 10^{-2}$) | Runtime (s)     |
> | ----------- | ------------------------------ | ---------------------- | --------------- |
> | **RAPID**   | **56.77±0.01**                 | **2.64±0.49**          | **1.07 ± 0.00** |
> | PID         | 56.76±0.01                     | 5.67±0.51              | 3.56 ± 0.01     |
> | MC-5        | 30.80±0.83                     | 13.01±0.35             | 5.81±0.51       |
> | MC-10       | 32.73±0.06                     | 10.78±0.44             | 13.46±1.14      |
> | Degree      | 54.28±0.01                     | --                     | 0.00± 0.00      |
> | Eigenvector | 29.78±0.00                     | --                     | 0.33± 0.00      |
> | PageRank    | 54.28±0.01                     | --                     | 0.18± 0.01      |
> | Betweenness | 47.80±0.01                     | --                     | 2.77± 0.02      |
> | Closeness   | 29.18±0.06                     | --                     | 1.10± 0.01      |
>
> ---
> - w3: Not compared with Bayesian inference (e.g. approximate Bayesian computation (ABC) or MCMC).
> - **R w3**: Bayesian methods (e.g., ABC, MCMC) solve the inverse problem: estimating epidemic parameters from observed time-series data (daily cases, deaths, etc.). Our RAPID method solves the inference problem: given known epidemic parameters and contact network structure, efficiently computing node-level infection probabilities at convergence. These are fundamentally different computational tasks with different inputs (time-series vs. network structure) and outputs (parameter estimates vs. node probabilities).
>
> ---
> - q1: The Monte Carlo simulations have stochasticity coming from random sampling of the initial infection node set.
> - **R1**: No, the stochasticity in MC simulations comes from random transmission and recovery events during propagation, **not** from sampling of the initial infected nodes. In `simulations/mc.py`, both `run_sir_simulation` and `run_monte_carlo_simulations` use a **fixed set** of initial infected node indices as inputs. The `test_sir_simulation` function, which samples initial seeds randomly, is only used for testing and is **not called by the main code**. The initial infected nodes are selected once in `main.py` (lines 207–208) and are kept the same across all methods (MC, PID, RAPID, centrality measures) to ensure a fair and direct comparison. This follows standard experimental protocol in epidemic modeling and matches our problem definition in the preliminaries ("given initial infected node set $\mathcal{I}_0$").
>
> Additionally, as shown in Appendix D.4, we tested different distributions for initial infected nodes, and the conclusions remained consistent, further demonstrating the efficiency and effectiveness of our RAPID algorithm.
>
> ---
> - q2: Can this framework be transferred into the Bayesian inference setting.
> - **R2**: Our method and Bayesian inference address different epidemic modeling problems. Our approach focuses on efficiently computing node-level marginal state distributions given known parameters and network structure. In contrast, Bayesian modeling estimates epidemic parameters from macroscopic observations like aggregate infection cases. When complete contact networks are available (e.g., from sensor data or mobility data), our RAPID method can **accelerate forward simulations for methods like ABC**, enabling faster parameter updates through efficient simulation-observation comparisons.
>
> ---
> - q3: Why is RAPID's accuracy higher than PID.
> - **R3**: This is an insightful question. As [7] notes in section "Dynamic Belief Propagation", when contact graphs contain many **short loops**, nodes within these loops cyclically influence each other's states over time, reducing the accuracy of the original PID algorithm. Our RAPID algorithm improves upon this through asynchronous max-heap iterations that **accelerate convergence** and selective updates that **avoid some short loops**, resulting in higher accuracy than PID.
>
> ---
> - q4: How accurate are 50-run Monte Carlo simulations.
> - **R4**: This is an important methodological concern. We provide the standard deviation of global mean infection probability for MC-50 across datasets (see table below; for hiv-Trans and carilion-Hospital, we use MC-1000 for higher reliability). For computational efficiency, we use MC-50 as baseline on large networks, and report the standard deviation across independent simulations as error bars in Table 3 and Table 4, ensuring that stochasticity is fully accounted for and avoiding bias from single-simulation outliers. These standard deviations and error bars (e.g. 1.27±0.47 for RAPID on hiv-Trans in Table 3) are consistently one to two orders of magnitude smaller than the effect sizes (e.g., MAE), confirming that MC-50 is sufficient for stable inference in practice. When propagation variance is higher (as in hiv-Trans and carilion-Hospital), we switch to MC-1000 to ensure robustness.
>
> | bitcoin-Alpha | email-Enron | email-EuAll | soc-Epinion | soc-Pokec | hiv-Trans* | carilion-Hospital* |
> | ------------- | ----------- | ----------- | ----------- | --------- | ---------- | ------------------ |
> | 0.001956      | 0.001144    | 0.000143    | 0.000090    | 0.000018 | 0.002580   | 0.004662           |
>
> **High-variance datasets use MC-1000 as baseline.*
>
>
>
> ---
> - q5: Paper Formatting Concerns.
> - **R5**: Thank you for pointing out. We will delete the repeated "variance of" in Line 113 and repeated "state" in Line 76. We will also change the title of Table 2 to "Dataset Statistics".
>
> ---
> [1] Morris, Martina, and Richard Rothenberg. "Hiv transmission network metastudy project: An archive of data from eight network studies, 1988--2001." (2011).
>
> [2] He, Yinhan, et al. "Demystify Epidemic Containment in Directed Networks: Theory and Algorithms." Proceedings of the Eighteenth ACM International Conference on Web Search and Data Mining. 2025.
>
> [3] Ru, Xiaolei, et al. "Inferring patient zero on temporal networks via graph neural networks." Proceedings of the AAAI Conference on Artificial Intelligence. Vol. 37. No. 8. 2023.
>
> [4] Adhikari, Bijaya, et al. "Fast and near-optimal monitoring for healthcare acquired infection outbreaks." PLoS computational biology 15.9 (2019): e1007284.
>
> [5] Prakash, B. Aditya, et al. "Threshold conditions for arbitrary cascade models on arbitrary networks." Knowledge and information systems 33.3 (2012): 549-575.
>
> [6]Karrer, Brian, and Mark EJ Newman. "Message passing approach for general epidemic models." Physical Review E—Statistical, Nonlinear, and Soft Matter Physics 82.1 (2010): 016101.
>
> [7] Lokhov, Andrey Y., Marc Mézard, and Lenka Zdeborová. "Dynamic message-passing equations for models with unidirectional dynamics." Physical Review E 91.1 (2015): 012811.

---

> ### Author Response · Authors · 2025-08-08
>
> Dear Reviewer,
>
> I hope this message finds you well. As the discussion period is nearing its end with **less than one day remaining**, I wanted to ensure we have addressed all your concerns satisfactorily. If there are any additional points or feedback you'd like us to consider, please let us know. Your insights are valuable to us, and we're eager to address any remaining issues to improve our work.
>
> Thank you for your time and effort in reviewing our paper.
>
> Best regards,
> The authors of Submission 17683

---

> > ### Comment · Reviewer_7Sx7 · 2025-08-08
> > **Response to rebuttal**
> >
> > Dear Authors,
> >
> > I sincerely apologize for my late response and thank you for your detailed rebuttal, especially the added experiments.
> >
> > Many thanks for clarifying the difference between the inference problem and the inverse problem, this clears up a lot of the confusion that I had.
> >
> > The additional experiments and corresponding explanations are clear to me and the results clearly demonstrate the advantage of what is proposed in this work.
> >
> > The hospital network experiment is a nice additional example, this very much addresses my concerns regarding real world applications.
> >
> > Considering these, I am willing to improve my score to a favorable one.

---

### Official Review · Reviewer_MyzN · 2025-07-03

**Clarity:** 2
**Significance:** 3
**Originality:** 3
**Rating:** 5
**Confidence:** 3

**Summary:**

This manuscript proposed 3 theoretical analysis on propagation model, and an algorithm called RAPID (Residual Aware Propagation for Infection Dynamics) based on the theoretical analysis, to efficiently approximate disease spread over networks. It addresses the high computational cost of traditional Monte Carlo (MC) simulation-based approaches by proposing three theoretical contributions:
1. A variance lower bound for the MC estimator
2. A linear approximation of nonlinear epidemic dynamics
3. A convergence condition for non-reinfection epidemic models
Based on these theoretical insights, the authors build RAPID to estimate individual-level infection dynamics more efficiently. The algorithm is tested on multiple datasets and shows promising improvements in runtime and comparable accuracy. The supplementary material also discusses extension to the SEIR model and provides a hyperparameter sensitivity analysis.

**Questions:**

1. Could the authors provide more insight or empirical justification for the preheating phase? Like what is the theoretical rationale behind using this initialization step? How should p be selected? Will that be possible if the authors could conduct ablation study on the preheating phase’s impact?
2. I checked the code, seems like bitcoin-Alpha data is not under /data/.
3. In paper 5.1 Baselines, the authors list Network centrality measures, MC-based inference, and probability-based inference (PID), but in all results tables, there are MC-5, MC-10, MC-50, PID, so where is Network centrality measures and how it works?
4. In manuscript, why MC-50 showed in Table 4 but not in Table 3?

**Ethical Concerns:**

["NO or VERY MINOR ethics concerns only"]

**Final Justification:**

The authors rebuttal addressed my concerns, so I update the rating.

**Limitations:**

yes

**Quality:**

3

**Strengths And Weaknesses:**

Strengths:
1. The theoretical analysis looks solid.
2. The RAPID algorithm (and the baseline PID algorithm) is well-structured and clearly presented.
3. The authors conduct experiments across multiple datasets and include a detailed hyperparameter sensitivity discussion.
4. SEIR extension and propagation modeling on individual level add applicability and relevance.

Weaknesses:
1. Like authors have discussed, the model is limited to non-reinfection dynamics, which restricts its scope for diseases like COVID-19 or flu where reinfection is a factor.
2. The preheating phase is not fully justified. Though the authors conducted sensitivity study, there is no ablation study showing its impact, and to choose the proper value remain unclear to me.
3. Some baseline experiments results are not very clear, see questions 3 and 4.

---

> ### Author Rebuttal · Authors · 2025-07-30
>
> We sincerely appreciate your efforts to review our paper and provide valuable suggestions. Below we address each concern in detail.
>
> ---
> - w1: The model is limited to non-reinfection dynamics.
> - **R w1**: Our method targets non-reinfection epidemic models (SIR and variants), which represent a well-established and widely important problem setting in epidemic modeling. This framework naturally applies to many practical scenarios: acute outbreaks (SARS, MERS), high-fatality diseases (Ebola, Marburg), vaccine-preventable diseases (measles, MPox), and cases where immunity duration exceeds analysis time frames. Extensive prior work in network epidemiology focuses on this same model class [1,2], reflecting its fundamental importance.
>
> Our contribution achieves significant computational improvements for these established and practically relevant models. We plan to extend our algorithm to reinfection diseases in future work.
>
> ---
> - q1: Insight or empirical justification for the preheating phase.
> - **R1**: The preheating phase is designed to **ensure that the linear approximation error remains small** when the epidemic parameter $\beta$ is not sufficiently small. Theoretically, if both $\beta$ and $\gamma$ are proportionally reduced to very small values according to Remark 1 in the Appendix, the linear approximation becomes highly accurate and preheating is unnecessary. However, in our experiments with $\beta=1/18$ and moderately dense networks like email-Enron ($\bar{k}=10.02$), the product $\beta\bar{k}$ is not much less than 1, resulting in non-negligible linearization error. When further scaling of parameters is not practical, the preheating phase guarantees that $\max_{v\in\mathcal{V}} P_I^v\leq \delta$, so that $\beta\bar{k}\delta\ll1$ and the linear approximation remains valid. Preheating iterates rapidly as only a small fraction of nodes are initially affected. Therefore, we recommend choosing the number of preheating steps $p$ based on a target $\delta$. Our sensitivity analysis in Appendix D.5 explores $p$ values from 0 to 50 on email-Enron, effectively including both ablation ($p=0$) and varying preheating, providing a more comprehensive understanding than a simple ablation study.
>
> ---
> - q2: Bitcoin-Alpha data is not found under `/data/`.
> - **R2**: Sorry for the confusion. We uploaded bitcoin-Alpha under the original filename "soc-sign-bitcoinalpha". We did not upload soc-Pokec due to file size limits, but it can be downloaded from Stanford's SNAP Dataset (social networks section). Our code includes the necessary preprocessing logic for this dataset.
>
> ---
> - q3: Where is Network centrality measures and how it works.
> - **R3**: Figure 2 shows the Kendall-tau rank correlation and runtime trade-off between RAPID and **baselines including network centrality measures** against MC-50. We use node centrality rankings (e.g., degree, betweenness) as proxies for infection probability rankings, as high-centrality nodes are commonly considered high-risk in epidemic studies before [3,4].
>
> While centrality measures show some alignment with true infection probabilities in certain contexts, no single centrality measure consistently ranks nodes by infection probability. This limitation becomes more pronounced at higher initial infection ratios $\alpha$, as shown in Appendix D.3.
>
> ---
> - q4: Why MC-50 showed in Table 4 but not in Table 3.
> - **R4**: In our experimental setup, MC-50 serves as the ground truth for infection probabilities. Therefore, Table 3 shows MAE relative to this baseline, while Table 4 compares absolute runtime performance including MC-50.
>
> ---
> [1] Karrer, Brian, and Mark EJ Newman. "Message passing approach for general epidemic models." Physical Review E—Statistical, Nonlinear, and Soft Matter Physics 82.1 (2010): 016101.
>
> [2] Lokhov, Andrey Y., Marc Mézard, and Lenka Zdeborová. "Dynamic message-passing equations for models with unidirectional dynamics." Physical Review E 91.1 (2015): 012811.
>
> [3] Bell, David C., John S. Atkinson, and Jerry W. Carlson. "Centrality measures for disease transmission networks." Social networks 21.1 (1999): 1-21.
>
> [4] Cohen, Reuven, Shlomo Havlin, and Daniel Ben-Avraham. "Efficient immunization strategies for computer networks and populations." Physical review letters 91.24 (2003): 247901.

---

> > ### Comment · Reviewer_MyzN · 2025-08-08
> >
> > Thanks for your answers and sorry for the late reply. I think my concerns are addressed, and I will update the rating.

---

> ### Author Response · Authors · 2025-08-08
>
> Dear Reviewer,
>
> I hope this message finds you well. As the discussion period is nearing its end with **less than one day remaining**, I wanted to ensure we have addressed all your concerns satisfactorily. If there are any additional points or feedback you'd like us to consider, please let us know. Your insights are valuable to us, and we're eager to address any remaining issues to improve our work.
>
> Thank you for your time and effort in reviewing our paper.
>
> Best regards,
> The authors of Submission 17683

---

### Official Review · Reviewer_iUqy · 2025-07-03

**Clarity:** 3
**Significance:** 2
**Originality:** 3
**Rating:** 5
**Confidence:** 3

**Summary:**

Researchers use Monte Carlo simulation to estimate the probability of infection over networks given disease dynamics. However, these methods have a high computational cost. This paper suggests a creative new solution for approximating the probability of node infection quickly. The authors also specify  convergence-criterion for non-reinfection epidemic models and bound Monte Carlo estimation error in an epidemics-over-networks context.

**Questions:**

1. What are the most predictable weaknesses of the linear approximation used in RAPID?

2. How does one choose $\epsilon$?

3. In the complexity analysis, is it only possible for nodes to be activated once?

4. What is $\ell_{ij}$ under equation 5? (I may have missed the definition somewhere)

I think it would clearer to move the ideas in remark 1 of the appendix (under section B.2) to the main body somewhere.

**Ethical Concerns:**

["NO or VERY MINOR ethics concerns only"]

**Final Justification:**

My main concern was that "the error inherent to their theoretical simplifications is not well explained. It seems that they use a lot of approximations in Theorems 3.1 and 3.2, and it’s difficult to understand (especially in Theorem 3.1) when these will be more/less accurate." The authors' rebuttal addressed this concern and I now believe this it is a relevant and technically solid paper.

**Limitations:**

yes

**Paper Formatting Concerns:**

none.

**Quality:**

2

**Strengths And Weaknesses:**

Strengths: This paper is very clearly written. The authors setup their problem and define their contributions well. RAPID is a creative algorithm, and they demonstrate that it performs well empirically across many metrics and datasets. It seems like a novel, useful, and practical tool for the field. Additionally, their theoretical contributions are interesting and help motivate aspects of their algorithm.

Weaknesses: The error inherent to their theoretical simplifications is not well explained. It seems that they use a lot of approximations in Theorems 3.1 and 3.2, and it’s difficult to understand (especially in Theorem 3.1) when these will be more/less accurate. Additionally, it would be useful to understand which network/disease contexts would predictably result in RAPID experiencing problems relative to Monte Carlo estimation.

---

> ### Author Rebuttal · Authors · 2025-07-31
>
> We sincerely appreciate your efforts to review our paper and provide valuable suggestions. Below we address each concern in detail.
>
> ---
> - w1: The error inherent to their theoretical simplifications is not well explained.
> - **R w1**: The approximation in Theorem 3.1 mainly relies on the sparse graph assumption, while the linearization approximation in Theorem 3.2 is justified by Appendix Remark 1. The error inherent to our RAPID algorithm comes from two main sources: (1) the linearization approximation error when computing the propagation residual, and (2) the independent path assumption used during both the preheating step and state updates, which matches the error sources in Algorithm 1. As discussed in Remark 1 (Appendix), the linearization error can be made negligible by proportionally reducing the epidemic parameters $\beta$ and $\gamma$ to sufficiently small values when modeling continuous infection processes.
>
> The primary source of RAPID's approximation error is the independent path assumption. While this assumption is standard in network epidemiology analysis, it introduces errors from non-independent path interactions (e.g., network loops). As shown by [1], Algorithm 1 is exact on tree graphs and provides rigorous upper bounds for the number of infected individuals on arbitrary networks. [2] demonstrates that this approach typically yields accurate estimates on real sparse networks for a broad class of spreading dynamics. Therefore, RAPID’s approximation error is primarily influenced by graph sparsity and local structure: **it achieves high accuracy on commonly encountered sparse or locally tree-like networks**, while on denser networks it may overestimate marginal infection probabilities—but still provides a meaningful worst-case estimate, which is valuable for robust epidemic assessment. As shown in Table 3, RAPID achieves highly accurate results with MAE of 0.0127 and 0.0103 on the sparsest networks (hiv-Trans and email-EuAll), respectively.
>
> ---
> - q1: The most predictable weaknesses of the linear approximation used in RAPID.
> - **R1**: As discussed in Appendix Remark 1, epidemic parameters $\beta,\gamma$ need to be sufficiently small for the linear approximation to be accurate. If the parameters are large or the network is dense, RAPID may underestimate propagation residuals by neglecting higher-order effects, leading to missed activations for some nodes. The table below (bitcoin-Alpha) empirically demonstrates that smaller $\beta$ values result in lower MAE:
> | $\beta$ | $\gamma$ | MAE ($\times 10^{-2}$) |
> | ------- | -------- | ---------------------- |
> | 0.2     | 0.4      | 6.86±0.08              |
> | 0.1     | 0.2      | 5.72±0.10              |
> | 0.05    | 0.1      | 4.25±0.06              |
> | 0.01    | 0.02     | 3.01±0.11              |
>
> ---
> - q2: How to choose $\epsilon$.
> - **R2**: We have provided sensitivity analysis for both preheating steps $p$ and residual threshold $\epsilon$ in Appendix Section D.5 and Figure 10. For $\epsilon$ specifically, our analysis shows that large values skip informative updates while very small values have diminishing influence since residual scales are approximatedly bounded by $\bar{k}\beta$. In our experiments, $\epsilon=10^{-3}$ works well across datasets. Generally, $\epsilon$ should be 1-2 orders of magnitude smaller than $\bar{k}\beta$ to capture significant state changes while maintaining efficiency.
>
> ---
> - q3: Is it only possible for nodes to be activated once in the complexity analysis?
> - **R3**: No, in our algorithm, nodes can be activated and updated multiple times during asynchronous propagation. In our worst-case complexity analysis, we account for up to $\mathcal{O}(N)$ node activations. Here we present statistics on activation frequency versus node count for the bitcoin-Alpha dataset:
> | activate times | 0    | 1    | 2    | 3    | 4    | 5    |
> | -------------- | ---- | ---- | ---- | ---- | ---- | ---- |
> | Node count     | 35   | 3730 | 12   | 2    | 3    | 1    |
>
> We can see from the table that most nodes are activated once, making RAPID highly efficient.
>
> ---
> - q4: $\ell_{ij}$ under equation 5.
> - **R4**: Sorry for the confusion. $\ell_{ij}$ represents the path length between
> nodes $i$ and $j$. We will add this to Table 1 for clarity.
>
> ---
> - q5: Content arangement.
> - **R5**: We will move Remark 1 to precede the linearization analysis (i.e., starting from line 157 in the main text). Thank you for the feedback!
>
> ---
> [1] Karrer, Brian, and Mark EJ Newman. "Message passing approach for general epidemic models." Physical Review E—Statistical, Nonlinear, and Soft Matter Physics 82.1 (2010): 016101.
>
> [2] Lokhov, Andrey Y., Marc Mézard, and Lenka Zdeborová. "Dynamic message-passing equations for models with unidirectional dynamics." Physical Review E 91.1 (2015): 012811.

---

> ### Comment · Reviewer_iUqy · 2025-08-04
>
> The authors' responses were very helpful! Especially in relation to high accuracy in sparse or local tree-like structures. I have updated my score contingent on a more extensive discussion of the practical implications of theoretical simplifications being included in the paper.

---

> > ### Author Response · Authors · 2025-08-04
> >
> > Thank you for your efforts in reviewing our paper and for your constructive feedback during the rebuttal phase! Following your suggestions, we will include the extensive discussion of the practical implications of our theoretical simplifications from **R w1** before Theorem 3.1, Theorem 3.2, and after the algorithm design section (i.e., at Lines 117, 165, and 244, respectively) in the camera-ready version. We believe this will help readers more clearly understand the assumptions and applicability of our theory and algorithm. Thank you again for your valuable feedback!

---

### Note · Authors · 2025-08-11

Dear NeurIPS 2025 AC, SAC, and PC,

We express our sincere gratitude to all reviewers for their valuable feedback and constructive engagement during the rebuttal and discussion process. We are encouraged that multiple reviewers indicated our theoretical clarifications and additional experiments addressed their concerns. We appreciate the reviewers’ recognition of the strengths of our work, including **clear presentation** (Reviewers iUqy, 7Sx7, wj6a), **technical soundness** (Reviewers MyzN, wj6a), **novelty and originality** (Reviewers iUqy, MyzN, 7Sx7), and **thorough empirical evaluations** (Reviewers MyzN, 7Sx7, wj6a).

One major concern was **practical implications of our theoretical simplifications** (Reviewers iUqy, wj6a). We addressed this by clarifying literature connections and providing Table 3 analysis, demonstrating that our **RAPID** algorithm achieves high accuracy on sparse and locally tree-like graphs. Another concern was **real-world applicability** (Reviewers 7Sx7, wj6a). We clarified that our experiments include real epidemic dataset *hiv-Trans* and other widely used datasets. We added a hospital contact network from Carilion Hospital, Virginia, with consistent results.

We will further refine the paper in the camera-ready version to improve accessibility for a broader audience. Specifically:
1. Include an extensive discussion of the practical implications (R w1 with Reviewer iUqy) of our theoretical simplifications before Theorems 3.1 and 3.2, and after the algorithm design section (Lines 117, 165, 244);
2. Incorporate the *Carilion-Hospital* results into Figures 2 and 4, and Tables 2–4;
3. Expand explanation for why RAPID improves PID accuracy in Table 3 discussion (starting at Line 297, R3 with Reviewer 7Sx7).

We respect the concern on NeurIPS scope alignment (Reviewer wj6a). We clarified that our work connects to **Theory**, **Probabilistic Methods**, and **Machine Learning for Science**, aligning with NeurIPS 2025 Call for Papers. Reviewers MyzN and 7Sx7 also recognized our work's significance.

Finally, we are grateful that several reviewers highlighted the quality and potential impact of our work. We believe our contributions advance scalable inference for structured stochastic processes, bridging theoretical insight with practical utility. We respectfully submit that these qualities make it a valuable addition to the NeurIPS community and a strong candidate for acceptance.

Best regards,

The authors of Submission 17683

---

### Decision · Program_Chairs · 2025-09-17

**Decision:**

Accept (poster)

**Comment:**

This paper is recommended for acceptance due to its technical soundness, clear presentation, and the novelty of its proposed RAPID algorithm for scalable epidemic inference. The authors provided a thorough rebuttal that successfully addressed the reviewers' primary concerns, notably by clarifying the practical implications of their theoretical assumptions and by adding a new real-world hospital contact network dataset to strengthen the empirical validation. While one reviewer raised a valid point regarding the paper's scope, the work's contributions to scalable probabilistic inference on networks align well with NeurIPS' focus on theory, probabilistic methods, and machine learning for science.